# Cryo-EM analyses of dimerized spliceosomes provide new insights into the functions of B complex proteins

Zhenwei Zhang[1,5,6], Vinay Kumar [ID][2,6], Olexandr Dybkov [ID][2,3], Cindy L Will[2], Henning Urlaub[3,4], Holger Stark [ID][1✉] & Reinhard Lührmann [ID][2✉]

## Abstract

**The B complex is a key intermediate stage of spliceosome assembly. To improve the structural resolution of monomeric, human spliceosomal B (hB) complexes and thereby generate a more comprehensive hB molecular model, we determined the cryo-EM structure of B complex dimers formed in the presence of ATP $\gamma$S. The enhanced resolution of these complexes allows a finer molecular dissection of how the 5′ splice site (5′ss) is recognized in hB, and new insights into molecular interactions of FBP21, SNU23 and PRP38 with the U6/5′ss helix and with each other. It also reveals that SMU1 and RED are present as a heterotetrameric complex and are located at the interface of the B dimer protomers. We further show that MFAP1 and UBL5 form a 5′ exon binding channel in hB, and elucidate the molecular contacts stabilizing the 5′ exon at this stage. Our studies thus yield more accurate models of protein and RNA components of hB complexes. They further allow the localization of additional proteins and protein domains (such as SF3B6, BUD31 and TCERG1) whose position was not previously known, thereby uncovering new functions for B-specific and other hB proteins during pre-mRNA splicing.**

**Keywords** B complex; Cryo-EM; Higher Order Spliceosomal Complexes; pre-mRNA Splicing; Spliceosome
**Subject Categories** RNA Biology; Structural Biology

## Introduction

Spliceosome assembly is initiated by the interaction of the U1 snRNP followed by U2 snRNP with the 5′ss and branch site (BS), respectively, of the pre-mRNA intron to be spliced out (Kastner et al, 2019; Plaschka et al, 2019; Yan et al, 2019). This generates the spliceosomal A complex that is subsequently converted into a pre-B complex upon initial association of the U4/U6.U5 tri-snRNP, which interacts in part by formation of U2/U6 helix II (Boesler

et al, 2016). RNP rearrangements driven by the RNA helicase PRP28 lead to disruption of the U1/5′ss interaction (Chen et al, 2001; Staley and Guthrie, 1999), and stable integration of the tri-snRNP, converting the pre-B complex into the spliceosomal B complex. In the latter the U6 snRNA ACAGA box forms base pairs with intron nucleotides (nts) of the 5′ss, forming the U6/5′ss helix. U6 nts directly upstream interact with additional intron nts downstream of the 5′ss, leading to an extended (ext) U6/5′ss helix. The transformation of the pre-catalytic B complex into a catalytically active spliceosome occurs in multiple steps with first the pre-B^act-1 and pre-B^act-2 complexes formed (Townsend et al, 2020), followed by the B^act and B* complex, the latter of which catalyzes the first step of the pre-mRNA splicing reaction. The activation process is initiated by the action of the RNA helicase BRR2, which disrupts the U4/U6 snRNA base pairing interaction, leading to U4 release (Absmeier et al, 2016) (see also Appendix Table S1 for summary of B complex protein roles). This allows the subsequent formation of U2/U6 helices Ia and Ib, and the U6 ISL, which coordinate the $Mg^{+2}$ ions required for splicing catalysis (Fica et al, 2013).

The main body of the spliceosomal B complex is formed by the U4/U6.U5 tri-snRNP and other non-snRNP proteins, whereas its upper "head" domain contains the U2 snRNP (Bertram et al, 2017; Zhan et al, 2018). The latter is comprised of a 3′ and 5′ domain, which contain, among others, the SF3a and SF3b heteromeric protein complexes, respectively. In the U2 snRNP 5′ domain, the U2/BS helix is encompassed by the HEAT domain of the SF3B1 protein. Comparison of the structure of the isolated tri-snRNP or its structure in pre-B complexes with that in the human B complex revealed that most protein components of the human tri-snRNP undergo major rearrangements during B complex formation (Agafonov et al, 2016a; Bertram et al, 2017; Charenton et al, 2019; Zhan et al, 2018). For example, the large scaffold protein PRP8 adopts a half-closed conformation in the B complex, due to the movement of the PRP8 Large domain towards the PRP8 N-terminal domain (Appendix Fig. S1). BRR2's helicase domain, which consists of its N-terminal and C-terminal helicase cassettes, is translocated from its position close to the PRP8 reverse transcriptase-like (RT) domain in the pre-B complex to the PRP8 endonuclease-like (En) domain in the B complex (Appendix

[1]Department of Structural Dynamics, Max-Planck-Institute for Multidisciplinary Sciences, Am Fassberg 11, 37077 Göttingen, Germany. [2]Cellular Biochemistry, Max-Planck-Institute for Multidisciplinary Sciences, Am Fassberg 11, 37077 Göttingen, Germany. [3]Bioanalytical Mass Spectrometry, Max-Planck-Institute for Multidisciplinary Sciences, Am Fassberg 11, 37077 Göttingen, Germany. [4]Bioanalytics Group, Institute for Clinical Chemistry, University Medical Center Göttingen, Robert-Koch-Straße 40, 37075 Göttingen, Germany. [5]Present address: State Key Laboratory of Biotherapy, West China Hospital, Sichuan University, Chengdu, Sichuan, China. [6]These authors contributed equally: Zhenwei Zhang, Vinay Kumar. ✉E-mail: hstark1@mpinat.mpg.de; reinhard.luehrmann@mpinat.mpg.de

Fig. S1). At the same time, the U4 Sm core domain docks at the interface between the N- and C-terminal helicase cassettes of BRR2, and the central single-stranded region of U4 snRNA is bound by the RecA domains of BRR2's N-terminal helicase cassette. Concomitant with the large-scale translocation of BRR2, the PRP8 RNase H-like domain (RH), PRP6 N-terminal HAT (half of a tetratrico) repeats, and U4/U6 proteins, also undergo structural rearrangements during B complex formation (Bertram et al, 2017; Charenton et al, 2019; Zhan et al, 2018).

During the human pre-B to B complex transition, several B-specific proteins are recruited, including SMU1, RED, FBP21, SNU23, MFAP1, PRP38A, UBL5, WBP11, and PQBP1 (Boesler et al, 2016). These proteins are conserved in higher eukaryotes, but in the yeast *S. cerevisiae*, homologues of solely SNU23, PRP38, UBL5 and MFAP1 are found. The B-specific proteins are released in a stepwise manner during activation of the human spliceosome (Townsend et al, 2020), and they are largely absent in B$^{act}$ complexes, indicating that they function primarily during the B and/or pre-B$^{act}$ complex stages. Indeed, PRP38 has been shown to be required for the formation of the B$^{act}$ complex (Blanton et al, 1992; Schütze et al, 2016). Furthermore, previous cryo-EM studies indicated that several B-specific proteins play a role in positioning the U6/5′ss helix, negatively regulating BRR2 activity, and/or stabilizing the overall structure of the B complex (Bertram et al, 2017; Plaschka et al, 2017; Bai et al, 2018; Zhan et al, 2018). However, the precise functions of several of the B-specific proteins remain unclear.

In previously reported cryo-EM structures of the human B complex, peripheral regions of the complex, including those comprising the U2 snRNP, as well as many domains of the B-specific proteins, could not be well resolved, apparently due to their structural flexibility (Bertram et al, 2017; Zhan et al, 2018). The poor resolution in several regions of the B complex cryo-EM density prevents the localization of additional spliceosomal components, hindering the generation of a comprehensive molecular model, a problem that also applies to all other spliceosomal complexes analyzed to date by cryo-EM. Previous studies revealed that human B complexes accumulate in splicing extracts in the presence of ATPγS, and that these B complexes can form higher-order complexes that appeared to be dimers (Agafonov et al, 2016b). While the functional relevance of these dimers remains unclear, dimerization may lead to the stabilization of peripheral, and also other regions, of the B complex, allowing for improved resolution of its cryo-EM structure.

Here, we have determined the 3D structure of dimerized human B complexes by cryo-EM, leading to improved resolution of the monomeric human B complex structure. This allowed a more reliable, enhanced molecular dissection of how the 5′ss is recognized at the B complex stage, and provided new information about the molecular interactions of the B-specific proteins FBP21, SNU23 and PRP38 with the U6/5′ss helix and with each other, and thus additional insight into their functions during pre-mRNA splicing. Our hB complex structure also revealed that MFAP1 interacts with multiple B complex proteins and, together with UBL5, forms an exon binding channel that stabilizes the 3′ end of the 5′ exon, while SMU1 and RED form a heterotetrameric complex within the spliceosome that bridges the U2 snRNP and tri-snRNP. Our structure further shows that the spliceosomal proteins BUD31 and TCERG1 are recruited already at the B complex stage, and their

position in hB reveals the likely roles that they play at this stage of the spliceosome assembly process. The improved resolution of the U2 snRNP-containing region of hB allowed us to more precisely localize the intron and proteins comprising the U2 3′ domain, as well as map for the first time the position of the SF3B6 protein in hB, which suggests that it may facilitate U2/U6 helix II formation at the pre-B complex stage. Finally, we elucidate the molecular interfaces that mediate B complex dimerization, providing initial evidence that the 5′ and 3′ exons from both protomers and their associated proteins form structurally heterogeneous globules that play a key role in B complex dimerization. Taken together, the human B complex structure presented here, not only yields more accurate models of the protein and RNA components of hB, but also allows the localization for the first time of additional proteins/ protein domains at this stage, revealing previously unknown functions for several of the B-specific proteins, and also other spliceosomal proteins during spliceosome assembly.

## Results and discussion

### Cryo-EM of a human B complex dimer

To purify human spliceosomal B complexes, we performed splicing in HeLa nuclear extract in the presence of ATPγS, which stalls spliceosome assembly at the B complex stage (Agafonov et al, 2016b). Spliceosomes assembled on MINX pre-mRNA were affinity purified and fractionated by glycerol gradient centrifugation at low (75 mM) salt. Purified spliceosomes contain uncleaved pre-mRNA and stoichiometric amounts of the U2, U4, U5 and U6 snRNAs, as well as U1 snRNA (Fig. EV1A), indicating that the vast majority of them are stalled prior to the BRR2-mediated start of spliceosome activation, and thus are spliceosomal B complexes. Consistent with this, abundant proteins in these complexes include, among others, all of the U2 snRNP, U4/U6.U5 tri-snRNP and B-specific proteins (Appendix Table S2).

We determined the 3D structure of these affinity-purified spliceosomal complexes by performing single particle cryo-EM. The 3D reconstruction revealed a dimerized complex that consists of two spliceosomal B complexes that are aligned in a parallel manner (Fig. 1A). That is, U2 snRNP is located at the top, while the U5 Sm core of the tri-snRNP is located at the bottom of each of the protomers comprising the dimerized complex. The protomers contact each other via a centrally located interface and via two, poorly defined, large globular density elements at the top and bottom (Fig. 1A). In contrast to supraspliceosomes that are large (ca 200S) endogenous complexes that are isolated from cell nuclei and appear to consist of four spliceosomal subunits that form on a single pre-mRNA (Sebbag-Sznajder et al, 2020), the spliceosomal B complex dimers that we describe here are formed from two spliceosomes that each assemble in vitro on a separate pre-mRNA substrate, and sediment at a much lower S value (Agafonov et al, 2016b).

We subsequently performed focused classification and masked refinement, resulting in a resolution of the B complex 'monomer' of 2.9–3.5 Å throughout the U4/U6.U5 tri-snRNP (Fig. EV1B–G and Appendix Table S3). The B monomer structure, which appears to be the same for both protomers, exhibits significantly improved resolution compared to the previously reported hB complex

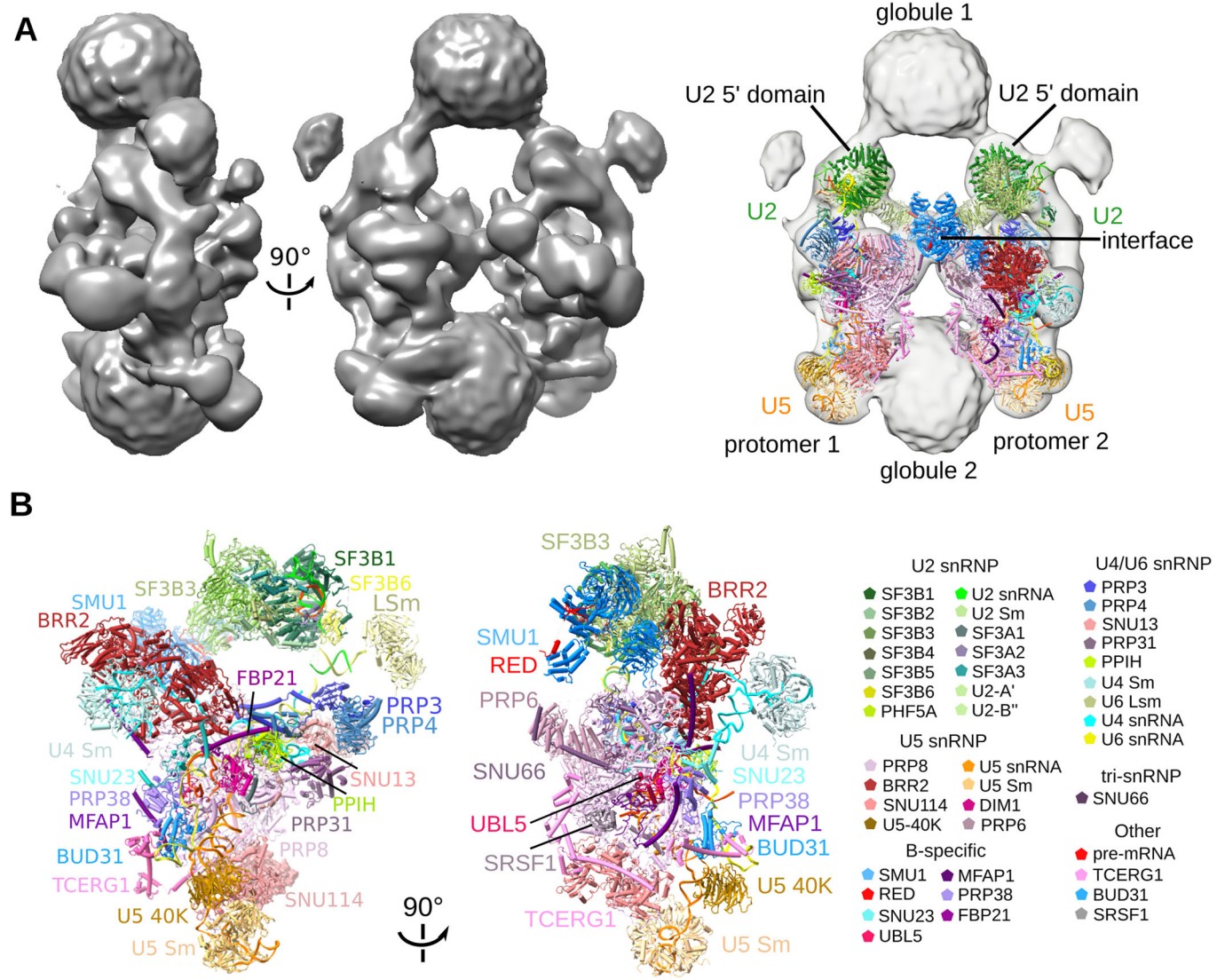

**Figure 1.   3D structure of dimerized, human B complexes.**

(A) Left and middle panels, different views of the EM density map of the human B complex dimers. Right, Fit of the B complex molecular model into each of the two halves of the dimer, which are connected by a middle interface and an upper and lower globule. (B) Two different views of the molecular architecture of the human B complex. At the right, summary of all modeled proteins and RNAs with color code. For protein gene names see Appendix Table S2.

structures (Bertram et al, 2017; Zhan et al, 2018). This allowed us to build more accurate models of the protein and RNA components, not only at its RNP core, but also at more peripheral regions of the tri-snRNP and U2 snRNP (Fig. 1B and Appendix Tables S4, S5). In addition, protein crosslinking coupled with mass spectrometry (CXMS)(Dataset EV1) allowed us to fit known protein structures or those predicted by AlphaFold into less well resolved density elements of the B complex, yielding the most comprehensive model of the human spliceosomal B complex to date (Fig. 1B and Appendix Tables S4, S5). Importantly, a comparison with the previously reported structures of the hB complex indicates that neither the presence of ATPγS nor B complex dimerization alters the overall structural organization of the hB protomers. That is, all major rearrangements in tri-snRNP components that accompany its stable integration into the B complex are also observed,

including, among others, formation of a half-closed conformation of PRP8 and translocation of BRR2.

## Improved molecular insight into 5′ss recognition at the B complex stage

The 2.9 Å resolution of core regions of our B complex structure allows more precise molecular insight into how nucleotides at/near the 5′ss of the pre-mRNA are recognized at the B complex stage. The 5′ exon nts −4 to −1 are base paired to U5 loop I nts U43 to U40 (Figs. 2A,B, EV1H and EV2A–D), whereby $G^{−1}$ forms a non-Watson-Crick base pair with U40, which is additionally stabilized by hydrogen bonding of K1306 of PRP8 to N7 of $G^{−1}$ (Figs. 2B–D and EV2E). The recognition of $G^{−1}$ by both U5 and PRP8, is consistent with previous studies indicating that PRP8 can functionally compensate for U5 loop

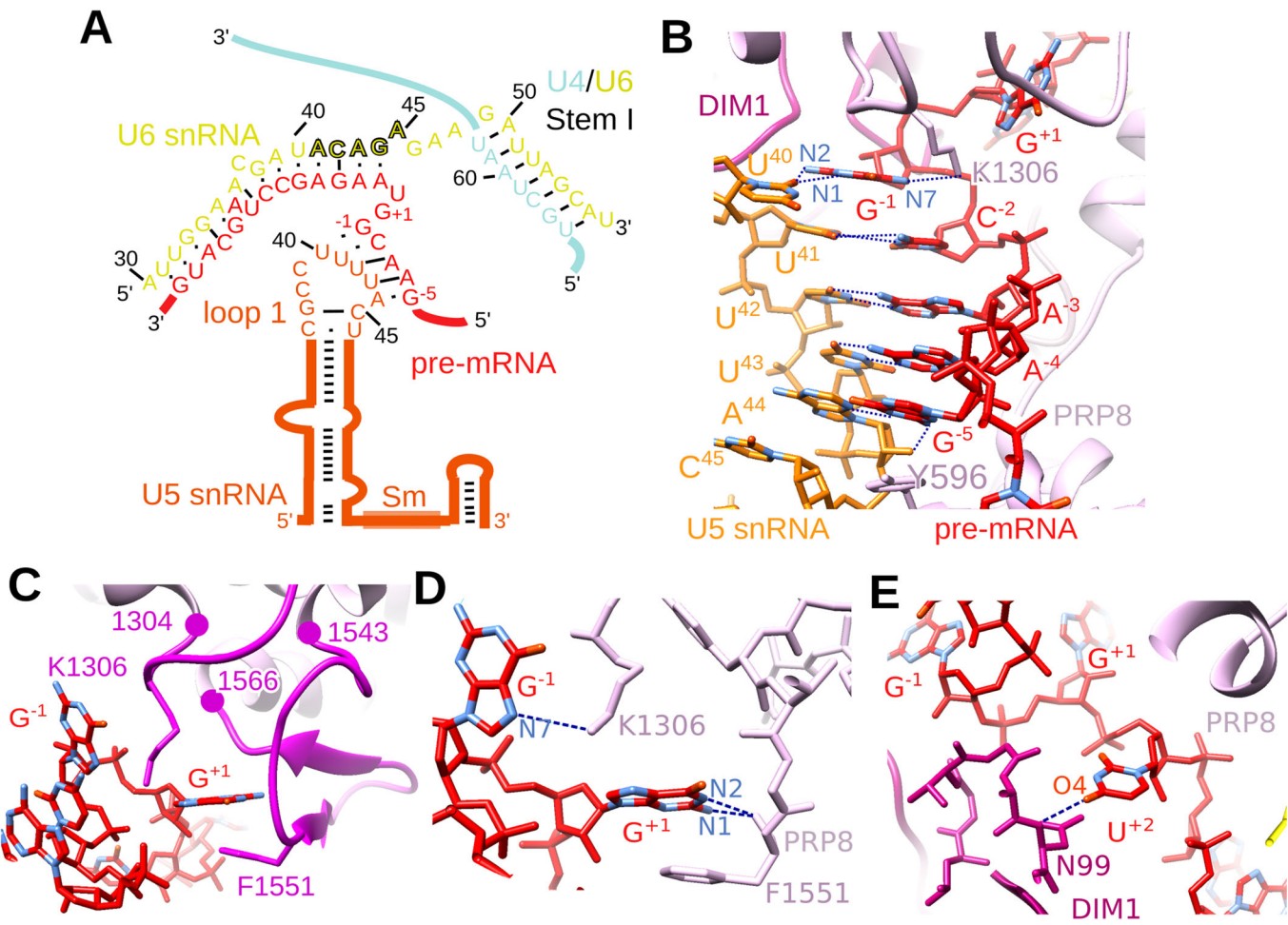

**Figure 2. Molecular recognition of nucleotides at/near the 5′ss by PRP8 and DIM1.**

(A) 2D schematic of the RNA network at/near the 5′ss in the hB complex. (B) Base pairing interactions between nucleotides of U5 snRNA loop 1 and nucleotides at the 3′ end of the 5′ exon. Dashed blue lines, hydrogen bonds. (C) G + 1 of the 5′ss is buried in a pocket formed by two loops of PRP8. (D) G-1 and G + 1 of the 5′ss are stabilized by PRP8 residues. (E) The backbone of DIM1 forms a hydrogen bond with U + 2 of the 5′ss.

1 nts during mammalian splicing (Ségault et al, 1999). $G^{-5}$ of the 5′ exon adopts two alternate conformations in the B complex, one in which it base pairs with U5 and one where its base is flipped out (Fig. EV2B,C). Our structure reveals that the 5′ terminal nts of the intron, i.e., the 5′ss dinucleotides $G^{+1}$ and $U^{+2}$, are recognized in hB exclusively by amino acids of PRP8 and DIM1, respectively, in an extended conformation, and thus are not contacted by RNA. $G^{+1}$ is buried inside a pocket formed by two loops of the PRP8 Linker (aa1304–1312 and 1543–1566) (Figs. 2C,D and EV2E). While the base of $G^{+1}$ stacks with the side chain of F1551 (Figs. 2C,E and EV2E), N1 and N2 of $G^{+1}$ form two hydrogen bonds with the backbone of F1551 (Fig. 2E). $U^{+2}$ is turned away from $G^{+1}$ and forms two hydrogen bonds with the backbone of DIM1, that is G97 (N3) and N99 (O4) (Figs. 2E and EV2F). It is interesting to note that the protein binding pocket accommodating the 5′ss nts in the B complex is first created upon the translocation of the PRP8 Large domain towards the PRP8 N-terminal domain. Thus, the formation of the half closed PRP8 conformation will likely precede the recognition of G + 1 and U + 2 by PRP8 and DIM1. In addition to accommodating U + 2 of the 5′ss, DIM1 also contacts U6 snRNA nts 47–48 via R127

and K125 (Fig. 3A,B). As these U6 nts stabilize the quasi-pseudoknot structure of U4 nts 63–68 that is found in the U4/U6.U5 tri-snRNP of the pre-B complex (Charenton et al, 2019), DIM1 may help to destabilize this pseudoknot during the pre-B to B complex transition. Interestingly, a G126D mutation in S. pombe DIM1, which could potentially hinder structural changes in the U4/U6 snRNA during B complex formation, leads to a defect in cell cycle progression (Berry and Gould, 1997), raising the possibility that this arises due to a splicing defect in a pre-mRNA that encodes a cell-cycle relevant protein. DIM1, together with PRP8, are displaced from the 5′ss GU dinucleotide during the transformation of the B complex into the $B^{act}$ complex, and the 5′ss is handed over to the RNF113A and SF3A2 proteins (reviewed in Kastner et al, 2019). The latter, as well as proteins that sequester the pre-mRNA branch site adenosine (the nucleophile for the first catalytic step of splicing), are then displaced during B* formation by the action of the helicase PRP2 (Bai et al, 2020; Schmitzova et al, 2023), enabling the first step of pre-mRNA splicing. Thus, prior to step 1, the 5′ss is sequestered by different proteins, preventing the former from premature nucleophilic attack.

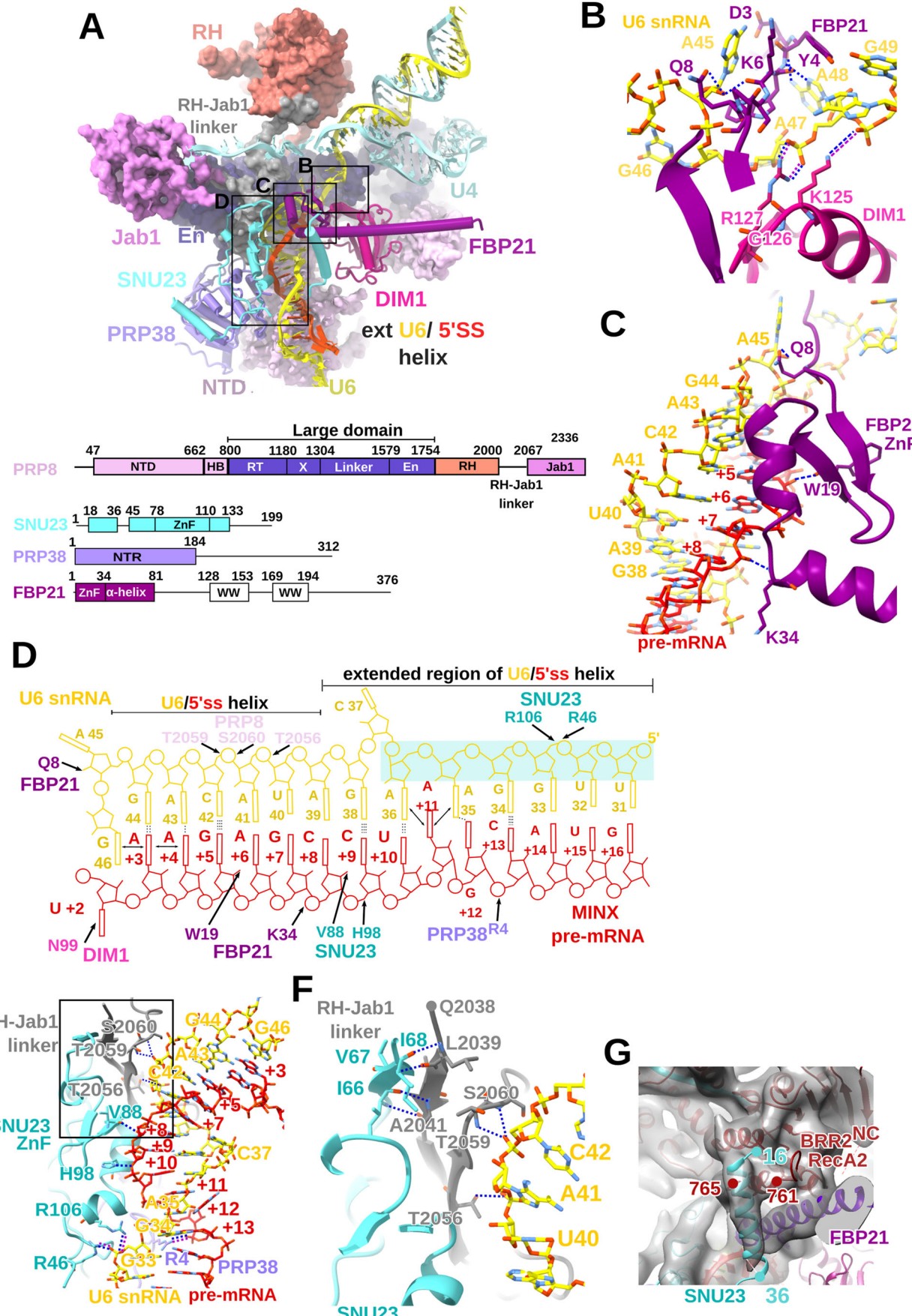

**Figure 3. Stabilization of the U6/5′ss helix by the B-specific proteins SNU23, FBP21, and PRP38.**

(A) Top, overview of the interaction of the U6/5′ss helix with the PRP8 N-terminal domain (PRP8$^{NTD}$), SNU23, FBP21 and PRP38. Boxes, regions expanded in panels (B), (C) and (D). Below, Schematic of the domain structures of PRP8, FBP21, SNU23 and PRP38. NTD N-terminal domain, HB helical bundle, RT reverse transcriptase-like, En endonuclease-like, RH RNase H-like, Jab1 Jab1/MPN domain, ZnF zinc finger, WW WW domain, NTR N-terminal region. (B) DIM1 and FBP21 contact U6 snRNA nts 47–49. (C) The zinc finger of FBP21 (FBP21$^{ZnF}$) interacts with intron nts +5 to +8 of the U6/5′ss helix. (D) Summary of protein contacts with the U6/5′ss helix nucleotides. (E) The zinc finger of SNU23 (SNU23$^{ZnF}$) interacts with intron nts that form part of the extended U6/5′ss helix. The boxed region is expanded in panel (F). (F) Interaction of a SNU23 loop (aa 45–75) with a β-hairpin (aa 2038–2067) of the PRP8 RH-Jab1 linker, which also interacts with U6 snRNA. (G) EM density fit of an SNU23 α-helix (aa 16–36) directly adjacent to α-helix (aa 761–765) of the BRR2 N-terminal helicase cassette (BRR2$^{NC}$).

## The zinc fingers of FBP21 and SNU23, together with PRP38, stabilize the U6/5′ss helix

The extended U6/5′ss helix runs along the PRP8 N-terminal domain and is contacted by the B-specific proteins FBP21, SNU23, and PRP38 (Bertram et al, 2017; Zhan et al, 2018) (Fig. 3A). The high resolution of our structure in this region reveals for the first time the molecular interactions of these three proteins with the U6/5′ss helix and with each other. The zinc finger domain of FBP21 contacts the U6/5′ss helix, interacting primarily with intron nts +6 to +8 (Fig. 3A,C,D). The SNU23 zinc finger is located directly adjacent to FBP21's zinc finger, and interacts with intron nts +9 and +10, as well as U6 nt 33 in the extended part of the U6/5′ss helical region (Fig. 3A,D–F). PRP38 also contacts the backbone of intron nt +13 (Fig. 3D,E). Both proteins interact primarily via charged amino acid side chains with nts of the U6/5′ss helix, stabilizing the extended helical region, which contains numerous non-canonical base pair interactions (Fig. 3D,E).

We could also de novo model seven amino acids located directly N-terminal of the FBP21 zinc finger. The density accommodating these residues was previously incorrectly assigned to U6 nts 46 and 47 (Zhan et al, 2018). Interestingly, the FBP21 N-terminus inserts deeply into the RNP core of the B complex and interacts with U6 snRNA nts 45 to 48 (Figs. 3B and EV2G), suggesting it assists DIM1 in destabilizing the pseudoknot structure of U4 nts 63–68 during the pre-B to B complex transition (see above). Alternatively, or in addition, the N-terminus of FBP21 may help to displace RBM42, as the binding of this region of FBP21 and the RBM42 RNA recognition motif (RRM) in the tri-snRNP, are mutually exclusive interactions (Charenton et al, 2019). As in previously reported hB complex structures (Bertram et al, 2017; Zhan et al, 2018), the long α-helix downstream of the FBP21 zinc finger is docked to PPIH (Fig. 1C), hindering access of the BRR2 N-terminal helicase cassette to U4/U6 stem I. Thus, FBP21, in conjunction with other spliceosomal proteins, aids in preventing the premature unwinding of the U4/U6 snRNA by BRR2, which first occurs during the transformation of the B complex into the B$^{act}$ complex.

SNU23 is recruited as a complex together with PRP38 to the B complex (Ulrich et al, 2016b), whereby the PRP38 α-helical N-terminal region bridges the endonuclease-like and N-terminal domains of PRP8. While in the previously reported crystal structure of truncated SNU23 and PRP38 only an extended α-helix of SNU23 (aa 118 to 133) was bound to PRP38 (Ulrich et al, 2016b), we can now model the entire interface between the ZnF of SNU23 and N-terminal region of PRP38 (Fig. EV3A,B). Guided by AlphaFold, we could also model a 31 aa long region (aa 45–75) upstream of the SNU23 zinc finger that forms a loop. This SNU23 loop interacts with a β hairpin structure of PRP8 (aa 2038–2067), which we could also model de novo and which covers a large part of the linker

between the PRP8 RNase H and Jab1 domains (aa 2016–2067). The latter linker (henceforth denoted the PRP8 RH-Jab1 linker), in complex with SNU23, also interacts with U6 snRNA nts that form the U6/5′ss helix in hB (Figs. 3D,F and EV3C). In the human pre-B complex, the PRP8 RH-Jab1 linker is flexible and could not be visualized (Charenton et al, 2019; Zhan et al, 2018). In contrast, it can be localized in hB where its ends now interact with the PRP8 endonuclease-like domain (Fig. 3E,F), suggesting that the RH-Jab1 linker is only stabilized when the PRP8 Jab1 domain is juxtaposed to SNU23 upon translocation of the PRP8 Jab1 bound BRR2 protein to the PRP8 endonuclease-like domain during B complex formation. Interestingly, guided by crosslinks (Dataset EV1), we could also place an N-terminal α-helix of SNU23 (comprised of amino acids 16 to 36) that was predicted by AlphaFold, into a density element that is connected directly to the RecA2 domain of the BRR2 N-terminal helicase cassette (Fig. 3G). Thus, SNU23, in complex with PRP38, may also help to stabilize the position of the BRR2 helicase domain in the hB complex.

## BUD31 and TCERG1 are recruited already at the B complex stage

TCERG1 is a metazoan-specific regulator of RNA polymerase II transcription elongation and pre-mRNA splicing (Montes et al, 2012; Montes et al, 2015) that was previously localized first in pre-B$^{act-1}$, but no longer in pre-B$^{act-2}$ or subsequently formed spliceosomal complexes (Townsend et al, 2020). Our data show that TCERG1 is already recruited to the B complex (Fig. 4 and Dataset EV1). While the C-terminal FF4-FF6 domains, as well as the long α-helix bridging FF3 and FF4, interact in both hB and pre-B$^{act-1}$ complexes with the PRP8 reverse transcriptase-like domain and SNU114 in essentially the same manner, the position of the FF1-FF3 domains is clearly different (Fig. 4A,B). That is, while in pre-B$^{act-1}$ FF1 interacts with MFAP1 and the RecA2 domain of the BRR2 N-terminal helicase cassette, in the B complex it interacts with BUD31 (Figs. 4B and EV3D). As a consequence, the FF1-FF3 region is located closer to the PRP8 N-terminal domain. Guided by crosslinks (Dataset EV1), we can also localize the RRM2 domain of the SR protein SRSF1 in a density element near TCERG1, very similar to its position in pre-B$^{act-1}$ (Figs. 4B and EV3D) (Townsend et al, 2020). TCERG1 is not present in pre-B complexes, where its binding would clash with that of PRP28 (Charenton et al, 2019). Indeed, numerous mutually exclusive protein interactions occur during the splicing process, and they ensure the sequential binding and release of many spliceosomal proteins, aiding the progression of not only spliceosome assembly but also its catalytic activation. The recruitment of TCERG1 during B complex formation is consistent with the idea that its main role is to stabilize part of the B complex that does not change substantially during the BRR2-mediated, structural

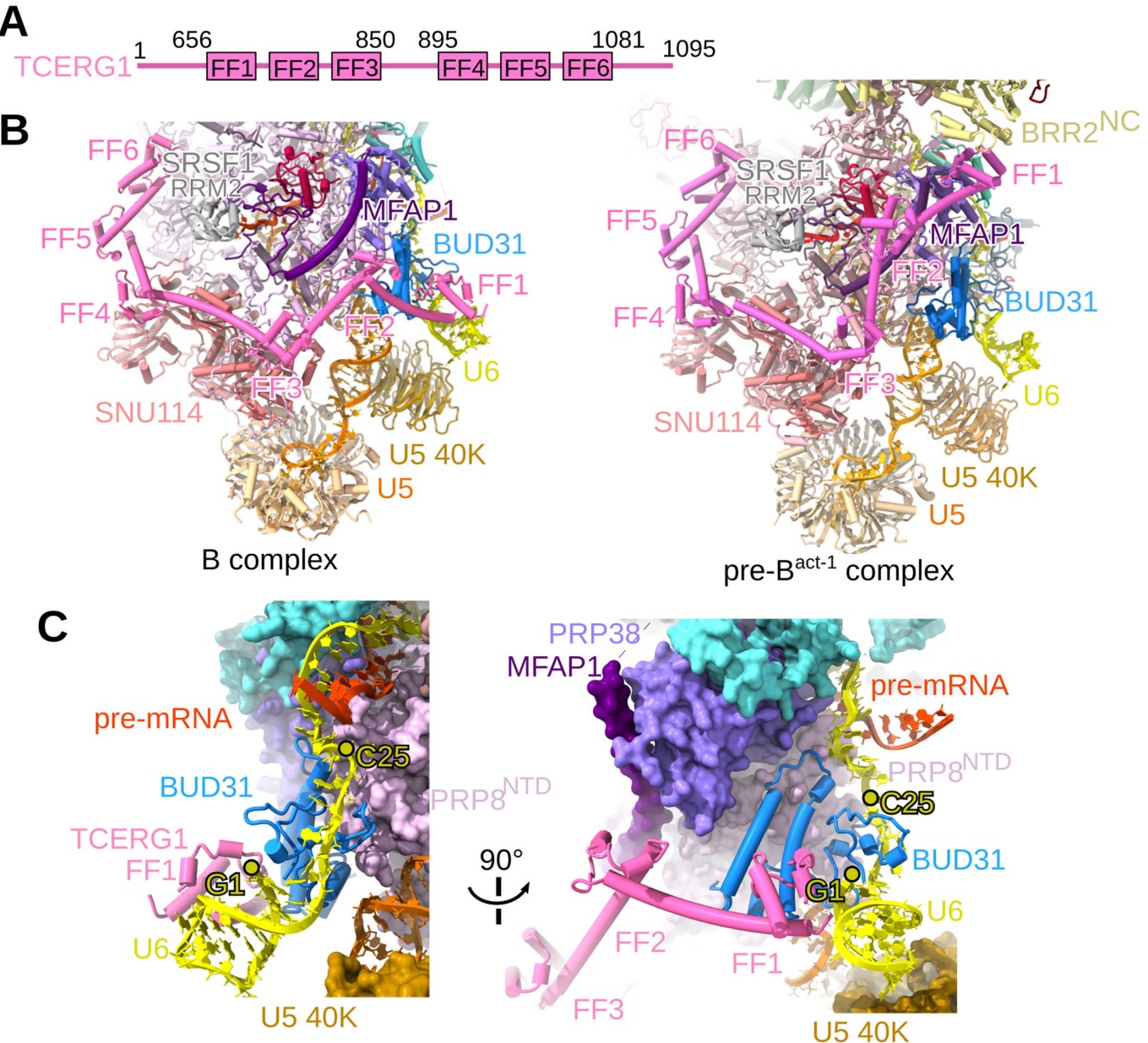

**Figure 4. TCERG1 and BUD31 are present in the hB complex.**

(A) Domain organization of TCERG1. (B) Molecular organization of the FF1-FF6 domains of TCERG1 in the human B (left) and pre-B$^{act-1}$ complexes (PDB 7ABG). C BUD31 is present in hB complexes and binds the 5′ end of the U6 snRNA.

rearrangements that occur at the early stages of spliceosome activation. Alternatively, or in addition, as TCERG1 interacts with RNAPII (Carty et al, 2000; Suñé et al, 1997), TCERG1 may facilitate communication between the transcription and splicing machineries at the B complex stage, and its binding could thus be an important regulatory checkpoint during co-transcriptional splicing. The presence at most stages of the splicing cycle of many additional spliceosomal proteins (including TCERG1) that are not present in the yeast *S. cerevisiae*, allows for additional, intermediate RNP conformations of the human spliceosome and the enhanced potential for additional regulatory checkpoints.

BUD31 is an evolutionarily conserved protein that was shown to bind to the PRP8 N-terminal domain and stabilize, in cooperation with RBM22, the 5′ end of U6 snRNA at PRP8 in pre-B$^{act}$ complexes (Townsend et al, 2020) and all subsequently formed spliceosomal complexes (Kastner et al, 2019). Our data reveal that BUD31 is already recruited at the B complex stage where, like in subsequently formed complexes, it stabilizes the 5′ end of U6 snRNA (Fig. 4B,C). As only low amounts of RBM22 are present in our purified hB complexes (Dataset EV1), BUD31 functions independently of RBM22 at this stage. However, in hB complexes BUD31 interacts with FF1 of TCERG1 (Fig. 4C),

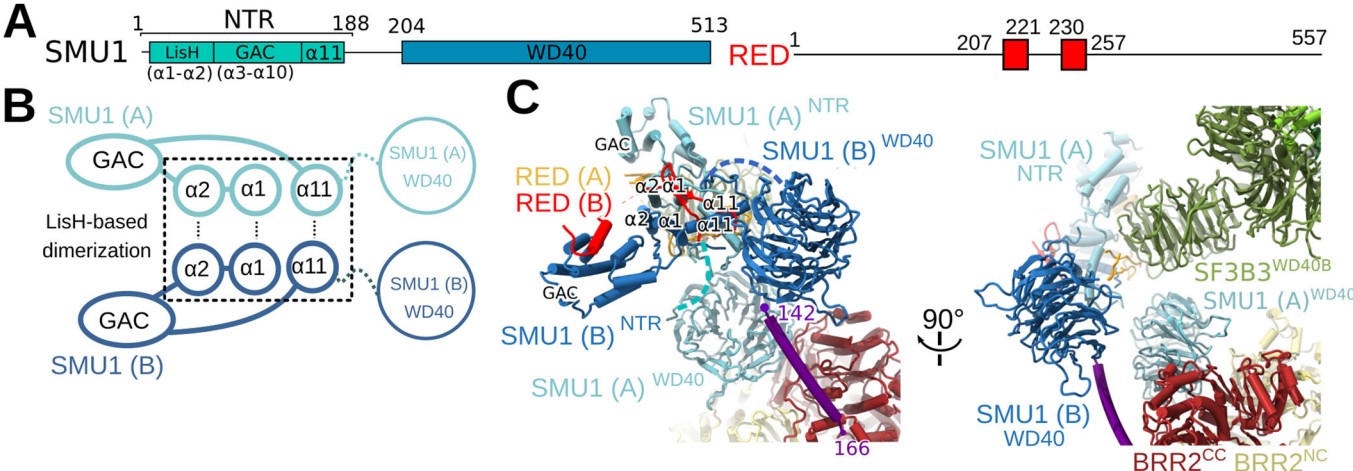

**Figure 5. A heterotetrameric SMU1-RED complex is present in the human B complex.**

(A) Domain organization of SMU1 and RED. NTR N-terminal region, LiSH Lissencephaly type 1-like homology, GAC globular α-helical core, WD40 β-propeller-like domain comprised of ca 40 amino acids that often contains a C-terminal tryptophan-aspartic acid (W-D) dipeptide. (B) Schematic of the organization of the SMU1 dimer. (C) Spatial organization of the SMU1-RED tetramer in the hB complex. Two different views are shown.

suggesting that the latter protein aids BUD31 in stabilizing the 5′ end of U6 prior to RBM22 recruitment.

## A heterotetrameric SMU1-RED complex in the human B complex

The B-specific protein SMU1 possesses an N-terminal, ca. 180 aa-long helical N-terminal region that contains LiSH and CTLH domains, which is connected via a short linker region to a C-terminal WD40 domain (Ulrich et al, 2016a) (Fig. 5A). Purified SMU1 dimerizes via its LiSH motif and a C-terminal α-helix, and the SMU1 dimer binds two copies of RED in solution or upon crystallization (Ashraf et al, 2019; Ulrich et al, 2016a) (Fig. 5B). In previous cryo-EM structures of hB complexes, only one copy of a SMU1-WD40 domain could be localized (Bertram et al, 2017; Zhan et al, 2018). Our structure shows for the first time that two copies of SMU1, which interact with each other via their N-terminal domains, and two copies of its interaction partner, RED, are present in the hB complex (Figs. 1B, 5C and EV4A). One of the SMU1 WD40 domains (henceforth termed SMU1A WD40) is located at the interface between the BRR2 N- and C-terminal helicase cassettes, and bridges BRR2 with the SF3B3 WD40B domain located in the U2 snRNP 5′ domain (Fig. 5C). The SF3B3 WD40B domain also shares a large interface with the SMU1 N-terminal dimerization domain. The WD40 domain of the second SMU1 copy (SMU1B WD40) is indirectly connected to the BRR2 C-terminal helicase cassette via MFAP1 α-helix[142-166] (Figs. 5C, 6A–C). Except for a short, middle region of RED (aa 207 to 257) that binds to the SMU1 N-terminal dimerization domain, the N- and C-terminal regions of RED are largely intrinsically unfolded and thus could not be precisely located in our structure. However, numerous crosslinks were detected between lysine residues in the N-terminal half of RED and those in SF3B1 and SF3B2, as well as with U6 LSm proteins, while lysines in the C-terminal region of RED were crosslinked to the PRP8 Large domain and NTD (Dataset EV1). Thus, RED, like SMU1, also plays a role in bridging

U2 with U5 proteins in the hB complex. As each B complex protomer contains two copies of RED, we cannot distinguish whether the crosslinks between RED and other B complex proteins are from one or both copies of the protein, and thus, whether the paths of both copies of RED are similar.

## MFAP1 bridges numerous B complex proteins and, with UBL5, forms a 5′ exon channel

The B-specific protein MFAP1 bridges SNU23 and PRP38 via a long acidic α-helix[271-313] (Bertram et al, 2017; Zhan et al, 2018) (Figs. 6A–C and EV4B). Guided by crosslinks and AlphaFold structure predictions, we could model in our hB complex structure, two additional, more N-terminally located MFAP1 α-helices that bind to the BRR2 N-terminal helicase cassette (MFAP1 α-helix[215-255]) or to SMU1B-WD40 (MFAP1 α-helix [142-166]) (Figs. 6A–C and EV4B). While we could not directly locate the MFAP1 N-terminal region, likely due to its flexibility, several crosslinks were detected between residues in this region of MFAP1 and the SF3B1 HEAT domain that clamps the U2/BS helix (Dataset EV1). Thus, the N-terminal ca. two thirds of the 439 aa-long MFAP1 protein span ~200 Å of the upper part of the B complex, and connect SMU1, BRR2 and the 5′ domain of U2 with the RNP core of the B complex.

We previously showed that MFAP1 and UBL5 form a 5′ exon channel in both the pre-B[act-1] and pre-B[act-2] complexes, and analysis of the EM density of a previously reported hB complex raised the possibility that this MFAP1/UBL5 exon channel might also be formed already in the B complex (Townsend et al, 2020). The high resolution of our B complex map allowed us to unambiguously model most of the C-terminal region of MFAP1 (aa 315-393) (Figs. 6D,E and EV4B). The latter forms a globular domain to which the ubiquitin-like B-specific protein UBL5 is docked. In a previously reported B complex structure, an α-helix of SNU66 was incorrectly placed in the density occupied by the C-terminal region of MFAP1 (Fig. EV4C) (Zhan et al, 2018). In our hB complex, the latter region of MFAP1 and UBL5 indeed form a continuous

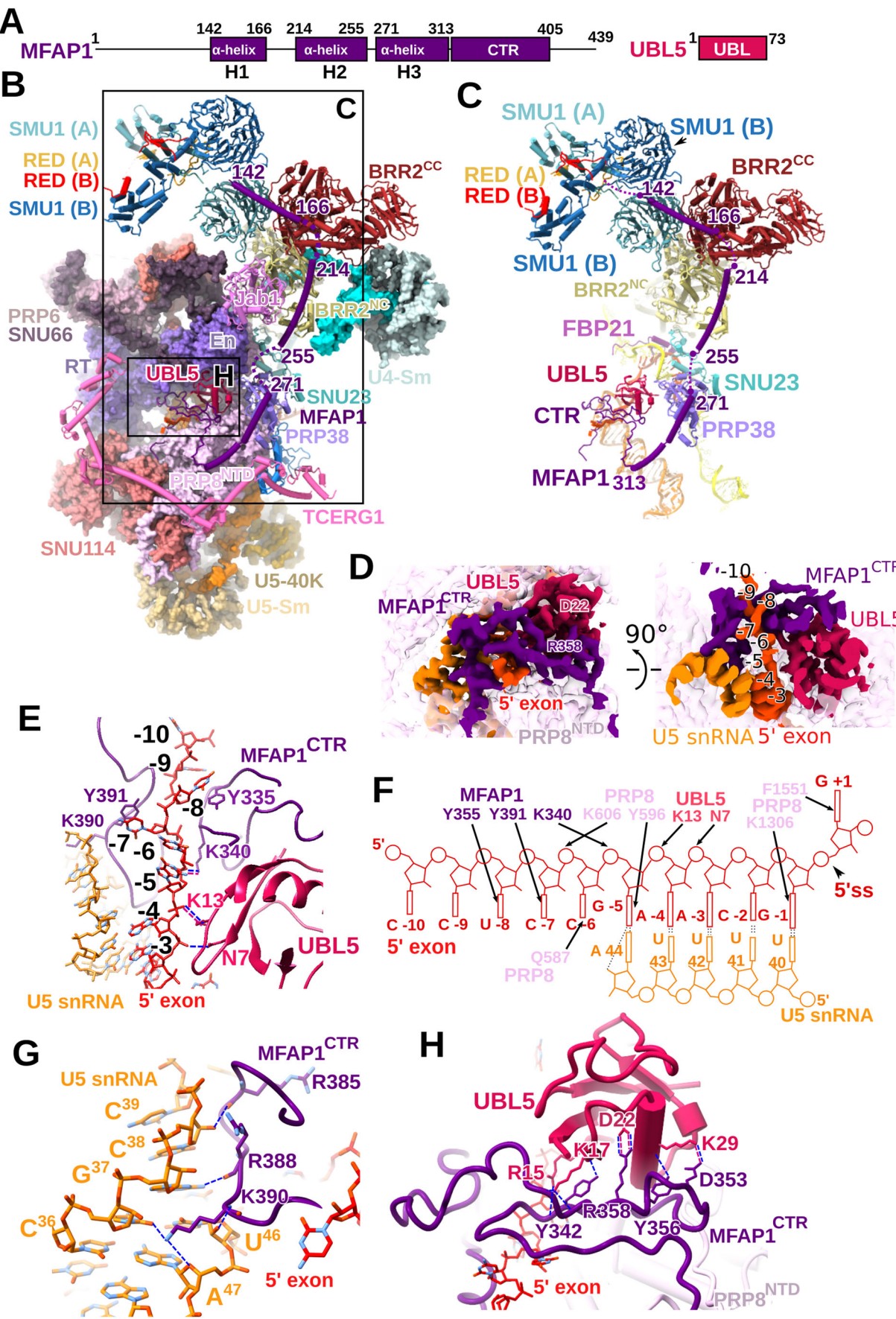

**Figure 6.  MFAP1 interacts with numerous proteins and, together with UBL5, forms part of a 5′ exon channel.**

(**A**) Domain organization of MFAP1 and UBL5. CTR C-terminal region, H helix, UBL ubiquitin-like. (**B, C**) A long α-helix (H3) of MFAP1 bridges SNU23 and PRP38, and helices H2 and H1 connect SMU1, BRR2 and the 5′ domain of U2 with the RNP core of the B complex. In panel (**B**) an overview of hB is shown where the boxed regions are expanded in panels (**C**) and (**H**), as indicated. (**D,E**) The MFAP1 C-terminal region (MFAP1^CTR) interacts with UBL5 and forms a channel that binds nucleotides at the 3′ end of the 5′ exon. Numbers indicate 5′ exon nucleotide positions relative to the 5′ss. In panel (**D**), colored EM density is shown, whereas a molecular model is shown in panel (**E**). (**F**) Summary of the molecular contacts of PRP8, the MFAP1 C-terminal region and UBL5 amino acids and U5 snRNA, with nucleotides at the 3′ end of the 5′ exon. (**G**) Amino acids in a loop of the MFAP1 C-terminal region interact with U5 loop 1 nucleotides. (**H**) Molecular contacts between the MFAP1 C-terminal region and UBL5. In panels (**G**) and (**H**), salt bridges and hydrogen bonds are indicated by violet and blue dashed lines, respectively.

channel in which both proteins interact with nucleotides at the 3′ end of the 5′ exon (Fig. 6C–F), and we can unambiguously identify the molecular contacts involving these proteins that stabilize the 5′ exon at this stage. Specifically, side chains of UBL5 N7 and K13 contact the phosphate groups of 5′ exon nts $A^{-3}$ and $A^{-4}$ that form base pairs with U5 loop 1 nts U42 and U43, respectively, stabilizing these base pairing interactions (Fig. 6E,F). MFAP1 then contacts the backbone of $G^{-5}$ via K340, while MFAP1 Y391 and Y335 stack with the bases of $U^{-7}$ and $U^{-8}$, respectively, guiding the exon towards the periphery of the exon channel (Fig. 6E,F). Interestingly, several amino acids within a loop of MFAP1, comprised of aa 385 to 390, interact with the backbone of U5 loop 1 nts C38 and G37, while K390 inserts into the loop, contacting C36 and A47 (Fig. 6G). It is likely that this MFAP1 region stabilizes the single-stranded part of U5 loop 1 and at the same time neutralizes the negative charges of the neighboring part of U5 loop 1 and nts at the 3′ end of the 5′ exon, thereby stabilizing the 5′ exon/U5 loop 1 base pairing interactions. This in turn will also stabilize the U6/5′ss helix. Deletion of HUB1 (the *S. pombe* homolog of human UBL5), as well as mutation of HUB1 D22 to an A, leads to aberrant alternative 5′ splice site usage in *S. pombe* (Mishra et al, 2011). Interestingly, in our B complex structure, UBL5 D22 is located at the interface of UBL5 and the MFAP1 C-terminal region, and interacts with MFAP1 R358 (Fig. 6H), suggesting that disruption (or distortion) of the UBL5/MFAP1 interface may lead to aberrant alternative splicing. Upon release of MFAP1 and UBL5 during B^act formation, the 5′ exon is handed over to SRRM2, which together with the PRP8 switch loop and CWC22, forms the exon binding channel in subsequently formed spliceosomal complexes (Kastner et al, 2019).

## SF3B6 is located at the SF3B1 C-terminal HEAT repeats near U2/U6 helix II

In previous B complex cryo-EM structures, the U2 snRNP was very flexible, allowing only rigid body docking of the entire U2 5′ domain from a hB^act complex structure into the hB EM density (Bertram et al, 2017; Zhan et al, 2018). In our B dimer complex, the U2 snRNP is more stable. This allows the fitting of individual structural elements of the U2 5′ domain into the corresponding densities of each protomer and indicates that in hB, the HEAT domain of SF3B1 has adopted a closed structure. In an in vitro assembled A-like complex, where the SF3B1 HEAT domain exhibits a half-closed conformation in which only the C-terminal HEAT repeats are docked onto the U2/BS helix (Tholen et al, 2022), SF3B6 could be localized at the C-terminal HEAT repeats. SF3B6 also interacts with the U2/BS helix two base pairs upstream of the bulged BS-A, consistent with it aiding in stabilizing the U2/BS helix. Guided by crosslinks, we can place in hB the RRM domain of SF3B6 at the same C-terminal region of the SF3B1 HEAT

domain (Fig. 7A–C). Moreover, our structure reveals that the SF3B6 RRM is located near U2/U6 helix II in hB (Fig. 7A–C). This raises the interesting possibility that, in addition to stabilizing the U2/BS helix, SF3B6 facilitates the formation of U2/U6 helix II by positioning the U2 snRNA strand upstream of the U2/BS helix in a manner conducive for base pairing with U6 snRNA during the docking of the tri-snRNP to the A complex.

SF3B6 is thus repositioned multiple times during the early stages of human spliceosome formation (Fig. 7D). In the isolated 17S U2 snRNP, the binding site at the C-terminal region of the SF3B1 HEAT domain is blocked by TAT-SF1 (Tholen et al, 2022; Zhang et al, 2020) and SF3B6 is flexibly attached to a short region of SF3B1 (aa 376–415) (Schellenberg et al, 2006) that is located N-terminal of its HEAT domain. During A complex formation, the SF3B6 RRM stably docks to the C-terminal HEAT repeats and remains there until B complex formation is completed. At the onset of BRR2-mediated activation of the B complex, SF3B6 is displaced by the PRP8 Large domain from its position at the C-terminal part of the SF3B1 HEAT domain, and is translocated to the outer region of the N-terminal HEAT repeats, close to SNIP1 in the pre-B^act-1 complex (Townsend et al, 2020). SF3B6 is thus another interesting example of how mutually exclusive protein–protein interactions in the splicing machinery drive the RNP structural dynamics of the spliceosome during its assembly and activation.

## Localization of the intron and proteins comprising the U2 3′ domain

The end of the U2/BS helix located at the C-terminal region of the SF3B1 HEAT domain appears to be connected with the U6/5′ss helix via a prominent, ca. 18 nm long density bridge, which is comprised of, among others, nucleotides of the MINX intron located upstream of the BS (Fig. EV5A), consistent with our previously reported B complex structure (Bertram et al, 2017). The most voluminous part of this intron bridge is sandwiched between the U4 Sm core and PPIH protein (Fig. EV5B). Guided by crosslinks we could dock the entire U2 3′ domain into a low pass filtered density map such that the SF3A2-β-sandwich domain is positioned into a globular density opposite the position of SF3B4 RRM1, thereby sandwiching intron nucleotides directly upstream of the BS (Fig. EV5B,C). We could also tentatively place RRM1 of hnRNP A1 close to the SF3A2 β-sandwich domain, while RRM2 of hnRNPA2 could be positioned near RRM2 of hnRNPA1 (Fig. EV5C). Moreover, K350 in the C-terminus of hnRNPA1 crosslinks to the BRR2 N-terminal helicase cassette and PPIH (Fig. EV5C). Thus, this region of the MINX intron is likely bound by hnRNP proteins, consistent with the broadness of the EM density bridge.

The high resolution of our structure also allowed us to localize part of the C-terminal region of SF3A1 that was modeled based on

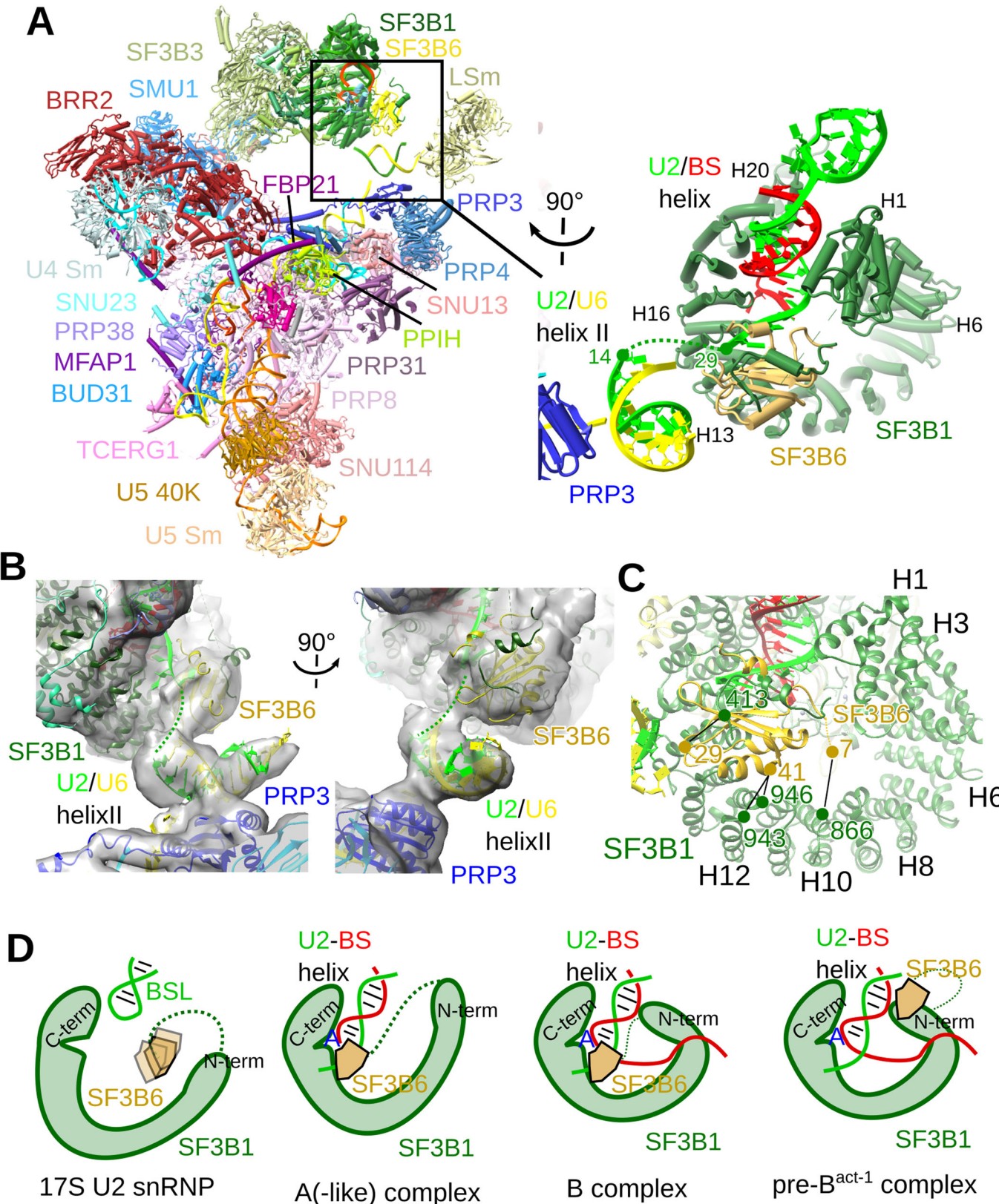

**Figure 7.  Localization of SF3B6 at the SF3B1 C-terminal HEAT repeats near U2/U6 helix II.**

(A) An overview of hB is shown on the left, where the boxed region is expanded and shown on the right. (B) Two views of the fit of SF3B6 into the hB complex EM density. (C) Protein crosslinks between SB3B1 and SF3B6 supporting the position of the latter in hB. Intermolecular crosslinks are depicted by a line, with the position of the crosslinked lysines indicated. (D) Cartoon summarizing the movement of SF3B6 during spliceosome assembly.

AlphaFold (Appendix Table S4). In hB, the region of SF3A1 comprised of aa 409–449 binds U4 snRNA SL1 (Fig. EV5D–H). A25 of the U4 snRNA is flipped out from SL1 and is contacted by numerous amino acids of SF3A1, and A25 thus tethers this part of SF3A1 (Fig. EV5H). SF3A1 (aa 409–449) further interacts with PRP4 and PPIH, and the more C-terminal α-helix (aa 454–489) of SF3A1 is docked at the well-structured interface formed between DIM1 and the PRP8 helical bundle (Fig. EV5F,G). In the previously published human pre-B complex (Charenton et al, 2019), aa 455–473 of SF3A1 were well resolved, and they are docked at the same position as in our B complex, suggesting that this short α-helix plays an important role in recruiting U2 to the tri-snRNP. As PPIH and the N-terminal region of PRP4 appear to be associated in a structurally flexible manner in the pre-B complex but not in hB, SF3A1 may also help to stabilize their position in the hB complex.

## Protein–protein interactions involving B-specific proteins contribute to B dimerization

The two protomers of the hB complex dimer are connected in the middle by a symmetrically organized interface involving the SMU1 dimerization domains and the N-terminal PRP6 HAT repeats, together with SNU66 α-helix[200–219] (Fig. 8A,B). That is, the SMU1 dimerization domain of hB protomer 1 is in close proximity to the N-terminal region of the PRP6 HAT repeats of hB protomer 2 and vice versa. SNU66 α-helix[200–219], which is bound to the "elbow-shaped" region in the N-terminal region of the PRP6 HAT repeats in each protomer, appears to bridge the latter with the N-terminal region of SMU1 (Fig. 8A,B). Formation of this dimer interface led to the finding that SMU1 and RED are present in hB in the form of a heterotetrameric complex, due to the improved resolution in this region. Thus, our studies indicate that the capture of higher-order complexes of spliceosomes indeed can help to improve the resolution of the EM density in some areas that are otherwise structurally, highly flexible.

## The 5′ and 3′ exons and associated proteins form the lower and upper globular domains, respectively

Both the upper and lower density elements that connect the two protomers of the B dimer are less structured and thus poorly resolved, compared to the middle interface of the dimer (Fig. 8A). An intriguing feature of the upper, globular density element (globule 1) is that it is connected to the U2 5′ domains of both protomers via thin density bridges that project out from the SF3B1 HEAT domain at the position where the 3′ end of the intron was previously to shown to exit the latter domain (Fig. 8C). This is consistent with the conclusion that these thin bridges are comprised of MINX pre-mRNA nucleotides in the vicinity of the 3′ss, and furthermore that the 3′ exons and associated proteins, including SR proteins, of both B protomers are located in globule 1 (Fig. 8A). The lower density element that connects the two protomers in the B dimer is positioned close to where the 5′ exon exits in both protomers, suggesting it is comprised of the 5′ exons and associated binding proteins (Fig. 8A,D).

Taken together, our data suggest that exons, presumably via a network of interactions involving various SR proteins that directly or indirectly contact the exon, have a tendency to interact with each other. This, in turn, could influence regulated splicing events involving alternative exon pairing. Consistent with the idea that different SR proteins interact transiently with the same exon, the globular domains of the hB dimer observed in 2D class averages by EM are very diffuse (Fig. EV1C), indicative of structural heterogeneity or the absence of a rigid RNP structure. SR proteins and other pre-mRNA splicing factors that have low-complexity, structurally disordered domains are prone to phase separation (Brangwynne et al, 2015; Greig et al, 2020), and RNP complexes have been shown to form so-called biomolecular condensates in the nucleoplasm (Banani et al, 2017). It is thus tempting to speculate that the diffuse globules that we observe in our hB dimers in vitro may contribute to bimolecular condensates containing exon RNA and, among others, SR proteins observed in vivo. Additional studies are, however, required to further characterize the exon-containing globules that mediate B complex dimerization and determine whether or not their formation provides any functional advantage during splicing.

# Methods

## MS2 affinity purification of human spliceosomal B complexes

HeLa S3 cells were obtained from the Helmholtz Zentrum für Infektionsforschung, Braunschweig and tested negative for mycoplasma. HeLa nuclear extracts were prepared according to (Dignam et al, 1983), and dialyzed twice for 2.5 h against 50 volumes of Roeder D buffer (20 mM HEPES-KOH, pH 7.9, 0.2 mM EDTA, pH 8.0, 1.5 mM MgCl$_2$, 100 mM KCl, 10% (v/v) glycerol, 0.5 mM DTT and 0.5 mM PMSF). Prior to use in the final reaction, the dialyzed HeLa nuclear extract was incubated at 30 °C for 25 min in the presence of 2 mM glucose. 10 nM m7G(5′)ppp(5′)G-capped MINX pre-mRNA containing three MS2 aptamers at its 3′ end was pre-incubated with 100 nM MS2-MBP fusion protein for 40 min on ice before addition to the splicing reaction. Splicing reactions were carried out for 90 min at 30 °C with 50% (v/v) nuclear extract in splicing buffer (1.5 mM MgCl$_2$, 65 mM KCl, 20 mM HEPES-KOH pH 7.9) containing 2 mM ATPγS. Splicing reactions were then chilled on ice for 10 min, centrifuged 15 min at 18,000 × g to remove aggregates and loaded onto a MBP Trap HP column (GE Healthcare). The column was washed with G-75 buffer (20 mM HEPES-KOH pH 7.9, 1.5 mM MgCl$_2$, 75 mM NaCl) and complexes were eluted with G-75 buffer containing 15 mM maltose. Eluted complexes were loaded onto a linear 10–30% (v/v) glycerol gradient prepared in G-75 buffer, centrifuged at 17,500 rpm for 18 h at 4 °C in a TST41.14 rotor (Thermo Fischer Scientific), and fractions were harvested from the bottom of the gradient. RNA from complexes in peak gradient fractions was separated on a denaturing 4–12%

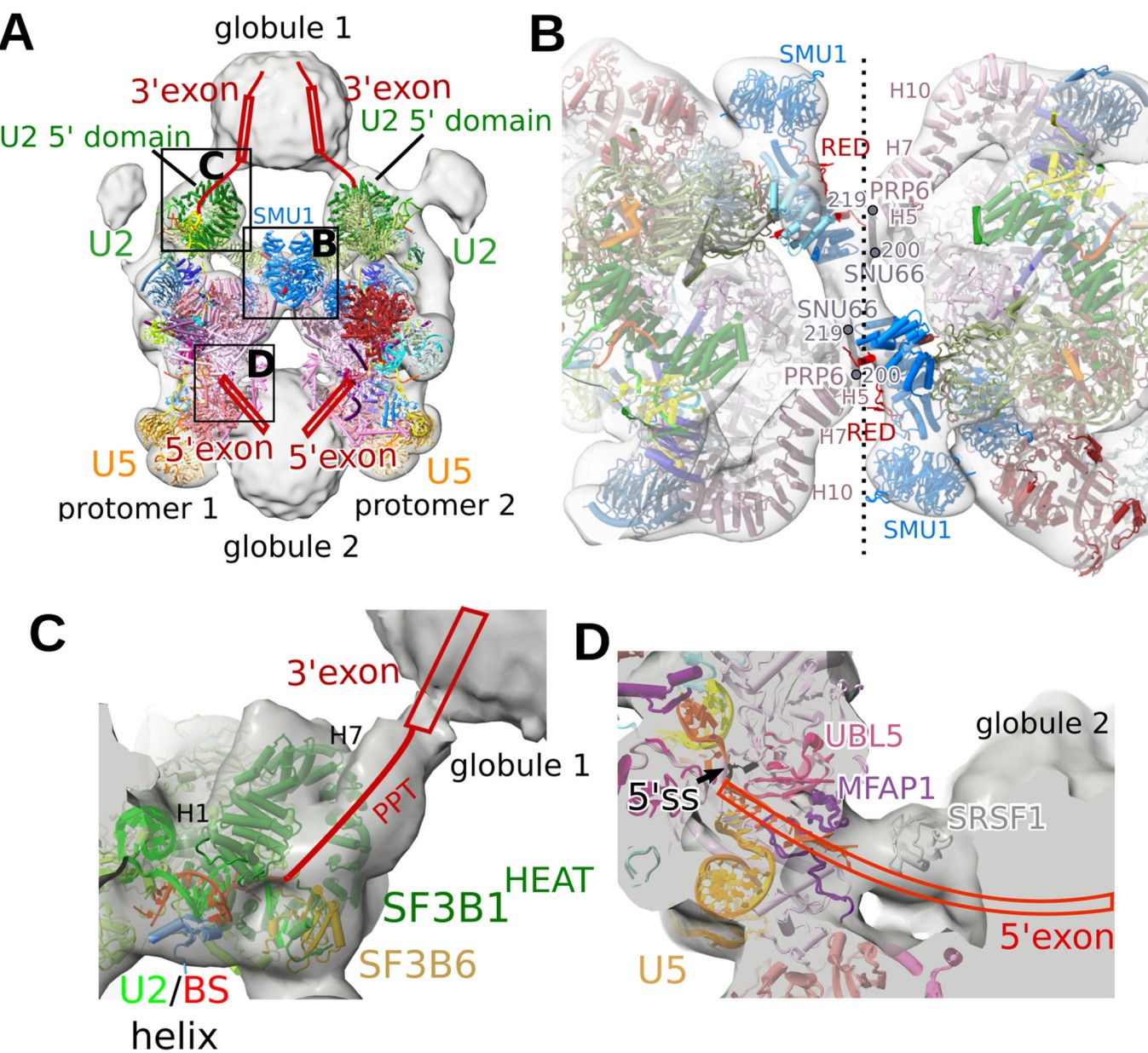

**Figure 8. The 5′ and 3′ exons and associated proteins form the lower and upper globular domains, respectively, of the hB dimer.**

(A, B) Protein–protein interactions involving B-specific proteins contribute to B complex dimerization. Panel (A), overview of the hB dimer, where the hB molecular models have been fit into the EM density and the predicted presence of the 3′ exon from each protomer is shown schematically in globule 1, and the 5′ exon from each protomer in globule 2. In panel (B), the boxed region in panel (A) has been expanded and rotated 90°. (C) EM density fit of the SF3B1 HEAT domain and the thin density element connecting it to globule 1. The likely path of the 3′ end of the intron and the 3′ exon is shown schematically. (D) EM density fit of MFAP1, UBL5 and SRSF1, with the probable path of the 5′ exon running into globule 2 shown schematically.

NuPAGE gel (Life technologies) and visualized by staining with SyBr gold (Thermo Fischer Scientific). For cryo-EM analyses, eluted complexes were subjected to gradient fixation (GRAFIX) and further processed as described below.

## Protein crosslinking of B dimers and crosslink identification

For protein–protein crosslinking and mass spectrometric experiments, spliceosomes were prepared as described above for cryo-EM but with

the following modifications: after MS2 affinity selection, eluted spliceosomal complexes were crosslinked with 350 μM BS3 for 35 min at 18 °C in a total volume of 1.6 ml, and subsequently subjected to glycerol gradient centrifugation at 17,000 rpm for 18 h at 4 °C in a TST41.14 rotor (Thermo Fischer Scientific) using a linear 10–30% (v/v) glycerol gradient. The three peak gradient fractions containing dimeric B complexes were pooled and pelleted by ultracentrifugation in a S100-AT4 rotor (Thermo Fisher Scientific). The pelleted, crosslinked dimeric B complexes (~7.5 pmol) were dissolved in 50 mM ammonium bicarbonate buffer, containing 4 M

urea, reduced with dithiothreitol, alkylated with iodoacetamide and, after diluting the urea to 1 M, in-solution digested with trypsin. Peptides were reverse-phase extracted using Sep-Pak Vac tC18 1cc cartridges (Waters) and fractionated by gel filtration on a Superdex Peptide PC3.2/30 column (GE Healthcare). Next, 50 µl fractions corresponding to an elution volume of 1.2–1.8 ml were analyzed in triplicate on a Thermo Scientific Orbitrap Fusion Lumos Tribrid mass spectrometer coupled to an Ultimate 3000 uHPLC (Thermo Scientific). The protein composition of the spliceosomal complexes was determined by a search with MASCOT 2.3.02 against a UniProt human reference proteome. Based on the MASCOT results, a restricted database was compiled and used for protein–protein crosslink identification by performing a search with pLink 2.3.9 (http://pfind.org/software/pLink/) according to the developer's recommendations (Chen et al, 2019). For simplicity, the crosslink score is represented as a negative value of the common logarithm of the original pLink score (i.e., Score = –log10(pLink Score). For model building, a maximum distance of 30 Å between the Cα atoms of the crosslinked lysines was allowed.

## EM sample preparation and image acquisition

Affinity-purified spliceosomal complexes were loaded onto a linear 10–30% (v/v) glycerol gradient prepared in G-75 buffer containing 0–0.1% glutaraldehyde (i.e., GRAFIX) and centrifuged at 17,500 rpm for 18 h at 4 °C in a TST41.14 rotor (Thermo Fischer Scientific). Fractions were then collected from the bottom of the gradient and quenched on ice by adding Tris-HCL, pH 7.5, to a final concentration of 120 mM. The peak fractions were combined, and then buffer-exchanged and concentrated using an Amicon 50 kDa cut-off unit. The concentrated sample was absorbed for 10 min to a thin layer carbon film that was subsequently attached to R2/2 UltrAuFoil grids (Quantifoil, Germany). 3.8 ml of double-distilled water was applied to the grids and excess water was blotted away by an FEI Vitrobot, using a blotting force of 11 and a blotting duration of 7.5 s, under conditions of 4 °C and 100% humidity, and subsequently vitrified by plunging into liquid ethane cooled to liquid nitrogen temperature. Cryo-EM grids were imaged in a Titan Krios (Thermo Fischer), equipped with a Cs corrector, operated at 300 kV on a Falcon III detector in linear mode at a calibrated pixel size of 1.16 Å at the specimen level. Cryo-EM images were taken using a total exposure time of 1.02 s (equivalent to 40 movie frames), with a total dose of 48 e-/Å2. No sample blinding was carried out.

## EM data processing

The recorded movie frames were aligned, dose-weighted, and summed by MotionCor 2.0 (Zheng et al, 2017). Defocus values were estimated by Gctf (Zhang, 2016). Particle picking was performed by crYOLO (Wagner et al, 2019). Approximately 1000 particles were manually picked from 50 micrographs and used to train a neural network model, which was then used for automated particle picking. All subsequent processing steps were performed using RELION 3.1 (http://www2.mrc-lmb.cam.ac.uk/relion/index.php/Main_Page) unless otherwise specified. In total, 25,718 micrographs were recorded, from which ~1 million particles were picked, extracted and pixel binned to 210 × 210 pixels (3× binned, pixel size of 3.48 Å). Several rounds of reference-free two-dimensional (2D) classification were performed, and the best appearing 2D classes containing 124,335 particles were selected for ab

initio reconstruction in cryoSPARC (Punjani et al, 2017). The ab initio reconstruction shows a dimerized complex with each protomer resembling a spliceosomal B complex, and the two protomers connected by poorly resolved globular densities and a middle interface. One protomer, as well as the unstable densities, were erased in Chimera (Pettersen et al, 2004), and the remaining density of the other protomer was low-pass filtered to 40 Å resolution to prevent model bias, which was then used for 3D classification in RELION 3.1. All of the ~1 million extracted particles were 3D classified into 5 classes, and the three good classes showing clear densities of B complexes were selected. Further 3D classification (no symmetry applied) showed that all of the retained particles are dimers that are formed in the same manner by two promoters aligned in a parallel manner. However, due to the continuous motion of the protomers within the dimer relative to one another, it was not possible to reconstruct both promoters simultaneously to high resolution.

To improve the resolution of the B complex protomer, the particles were recentered and re-extracted at the original pixel size in a 520 × 520 pixels box. The particles were 3D refined, and their alignment parameters were used for 3D classification without alignment (T = 40) with a soft mask around the tri-snRNP region. The best class containing 251,564 particles was selected, and two rounds of 3D refinement, CTF refinement, and Bayesian polishing were performed. In the final round of 3D refinement, soft masks around the tri-snRNP core (encompassing PRP8, the MINX pre-mRNA, U4/U6 stem I and II, U5 snRNA, PRP3, PRP4, SNU114, DIM1, PRP31, PRP6, SF3A1^409–489, SNU13, FBP21, SNU66, SNU23, the MFAP1 C-terminal region, UBL5, PRP38, BUD31, and TCERG1) and the BRR2 region were applied, yielding two 3D reconstructions with nominal resolutions of 3.1 Å and 4.2 Å, respectively. No symmetry was applied for any of the 3D reconstructions.

## Model building and refinement

Model building was carried out by docking cryo-EM, crystal, and AlphaFold2 structures into EM density and adjusting in COOT (Emsley and Cowtan, 2004). A complete list of modeled protein and RNA components, as well as their corresponding model templates is provided in Appendix Table S4. Briefly, the better resolved region of the previously published B complex model (PDB: 6AHD; including PRP8, U4/U6 stem I and II, U5 snRNA, SNU114, DIM1, SNU13, FBP21) was docked into the B complex protomer density as a rigid body, and individual components were manually adjusted, corrected, or extended in COOT to better fit the improved EM density. The MINX pre-mRNA (nts −10 to +21 relative to the 5′ss, corresponding to nts 49–79 of the MINX pre-mRNA) was de novo modeled into the EM density according to the corresponding sequence. The B-specific proteins MFAP1 (aa 271–315), SNU23, UBL5, and PRP38 were modeled based on AlphaFold2 predictions, then docked and adjusted into the EM density. The two N-terminal alpha helices of MFAP1 (aa 141–174 and aa 214–256) were truncated to polyalanine chains and docked into EM density separately. The model of SMU1 (N-terminal region) bound with RED was taken from the human co-crystal structure of SMU1-RED tetramer (PDB: 6Q8I) (Ashraf et al, 2019), truncated into polyalanine chains, and docked into EM density as one rigid body. The model of the SMU1 WD40 domain was taken from the human B complex (PDB: 6AHD), truncated to a polyalanine chain, and rigid body docked into EM density.

TCERG1 was modeled based on AlphaFold2 and truncated to a polyalanine chain, and its FF1-3 and FF4-6 were docked into the corresponding densities separately as two separate rigid bodies. BUD31 was modeled based on the B$^{act}$ structure (PDB: 6FF4) and manually adjusted into the density. The U2 5′ region (including U2 snRNA nts 33–65, SF3b complex, SF3A2 aa 41–85, and SF3A3 aa 393–462) and 3′ region (including U2 snRNA nts 97–184, the SF3a complex, U2 Sm, U2-A', and U2-B″) were taken from the human B$^{act}$ complex structure (PDB: 6FF7), truncated to polyalanine chains, docked as two separate rigid bodies without further adjustment. SF3B6 together with a piece of U2 snRNA (nts 29–32) was placed based on its position relative to the SF3B1 C-terminal HEAT domain in the A-like complex (PDB: 7Q4O) without further adjustment. Coordinates of the better resolved tri-snRNP region (encompassing PRP8, the MINX pre-mRNA, U4/U6 stem I and II, U5 snRNA, PRP3, PRP4, SNU114, DIM1, PRP31, PRP6, the C-terminal region of SF3A1, SNU13, FBP21, SNU66, SNU23, the C-terminal region of MFAP1, UBL5, PRP38, BUD31, and TCERG1) were refined in real space using PHENIX (Afonine et al, 2018). The lower-resolution parts are all modeled as polyalanine chains without further refinement.

## Data availability

The cryo-EM maps of the hB dimer, protomer, tri-snRNP and BRR2 regions are deposited in the Electron Microscopy Data Bank (EMDB) with the accession codes EMD-19063, EMD-18529, EMD-18225, and EMD-19062, respectively. The models of the tri-snRNP core region (8Q7N) and the entire hB complex (8QO9) of the human B complex protomer are deposited in the Protein Data Bank (PDB).

## Peer review information

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

## Acknowledgements

The authors would like to thank Thomas Conrad for HeLa cell production in a bioreactor, Hossein Kohansal for preparing HeLa nuclear extract, and Winfried Lendeckel, Monika Raabe und Ralf Pflanz for excellent technical assistance. We are grateful to Berthold Kastner for insightful discussions concerning the B complex structure. This work was supported by funding from the Max Planck Society (to RL). HU was supported by grants from the Deutsche Forschungsgemeinschaft (SFB860, project number 105286809, and SFB1565 project number 469281184). HS was supported by a grant from the Deutsche Forschungsgemeinschaft (SFB1565).

## Author contributions

**Zhenwei Zhang**: Formal analysis; Validation; Investigation; Methodology; Writing—review and editing. **Vinay Kumar**: Formal analysis; Investigation; Methodology. **Olexandr Dybkov**: Formal analysis; Investigation; Methodology. **Cindy L Will**: Supervision; Writing—original draft; Writing—review and editing. **Henning Urlaub**: Supervision; Funding acquisition. **Holger Stark**: Supervision; Funding acquisition. **Reinhard Lührmann**: Conceptualization; Supervision; Funding acquisition; Writing—original draft; Writing—review and editing.

## Funding

## Disclosure and competing interests statement

The authors declare no competing interests.

# Expanded View Figures

**Figure EV1. Cryo-EM and image-processing of the hB complex.**

(**A**) RNA composition of purified B complex dimers. Human (h) B complexes were affinity-purified and RNA from the peak gradient fractions was isolated, separated on a NuPAGE gel, and visualized by staining with SyBr gold. U1 snRNA, that is no longer base paired to the 5'ss of the MINX pre-mRNA is still present, together with U2, U4, U5 and U6 snRNA, and the MINX pre-mRNA. It is likely that, under our low salt purification conditions, U1 snRNP remains bound via protein–protein interactions (i.e., in the poorly resolved, globular EM densities). (**B**) Cryo-EM computation sorting scheme for the hB complexes. All major image-processing steps are depicted. (**C**) Representative cryo-EM 2D class averages of the human B complex dimers. (**D**) Local resolution estimation of the tri-snRNP core region of the B complex. (**E**) Orientation distribution plot for the particles contributing to the reconstruction of the tri-snRNP core region. (**F**) Fourier shell correlation (FSC) values for the listed parts of the B complex, indicate a resolution of 3.1 Å for the tri-snRNP core and 4.2 Å for the BRR2 region. (**G**) Map versus model FSC curves generated for the tri-snRNP core and BRR2 regions of hB using PHENIX mtriage. (**H**) Schematic of the RNA-RNA network in the hB complex. Red nucleotides are from the MINX pre-mRNA.

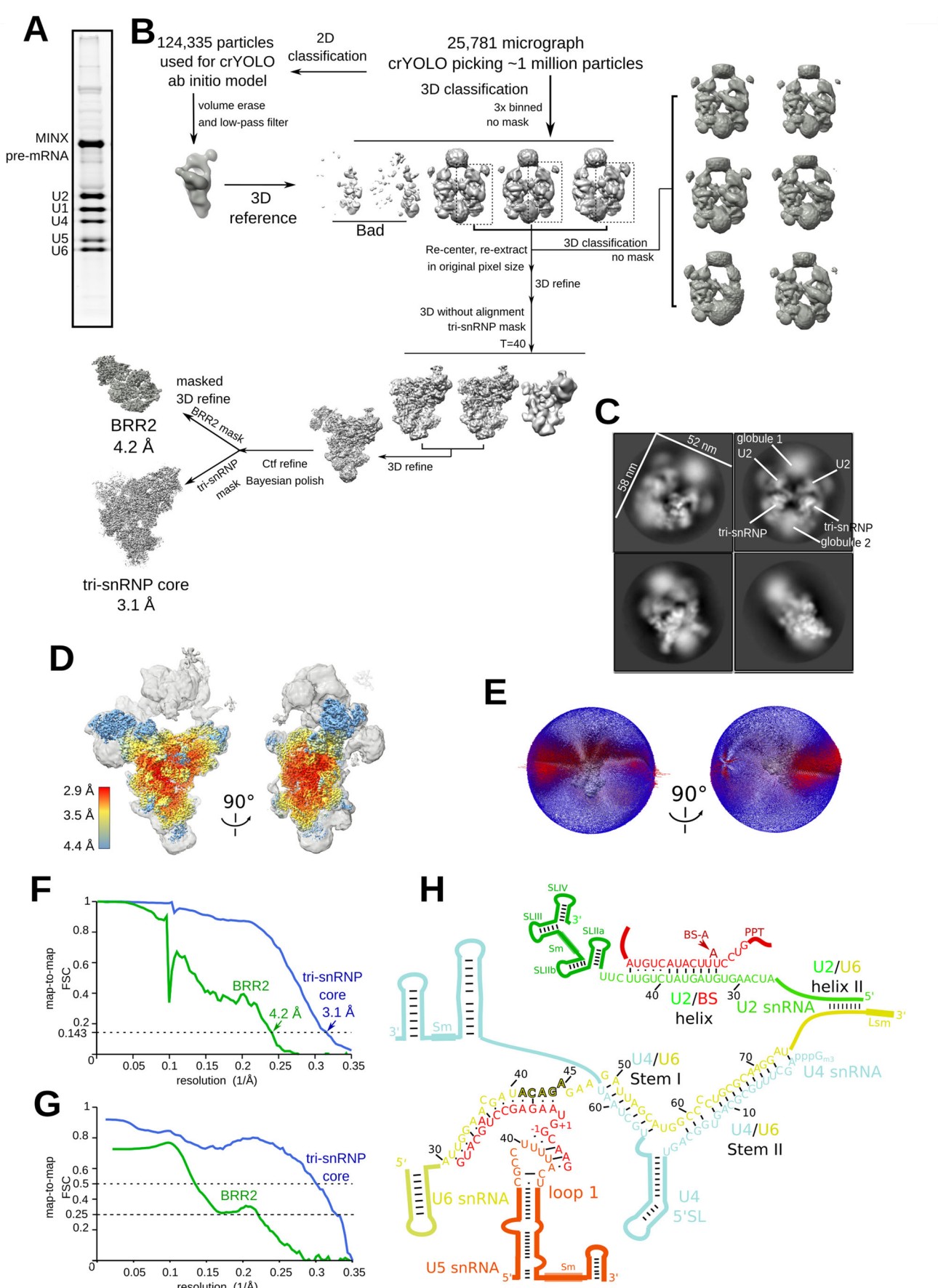

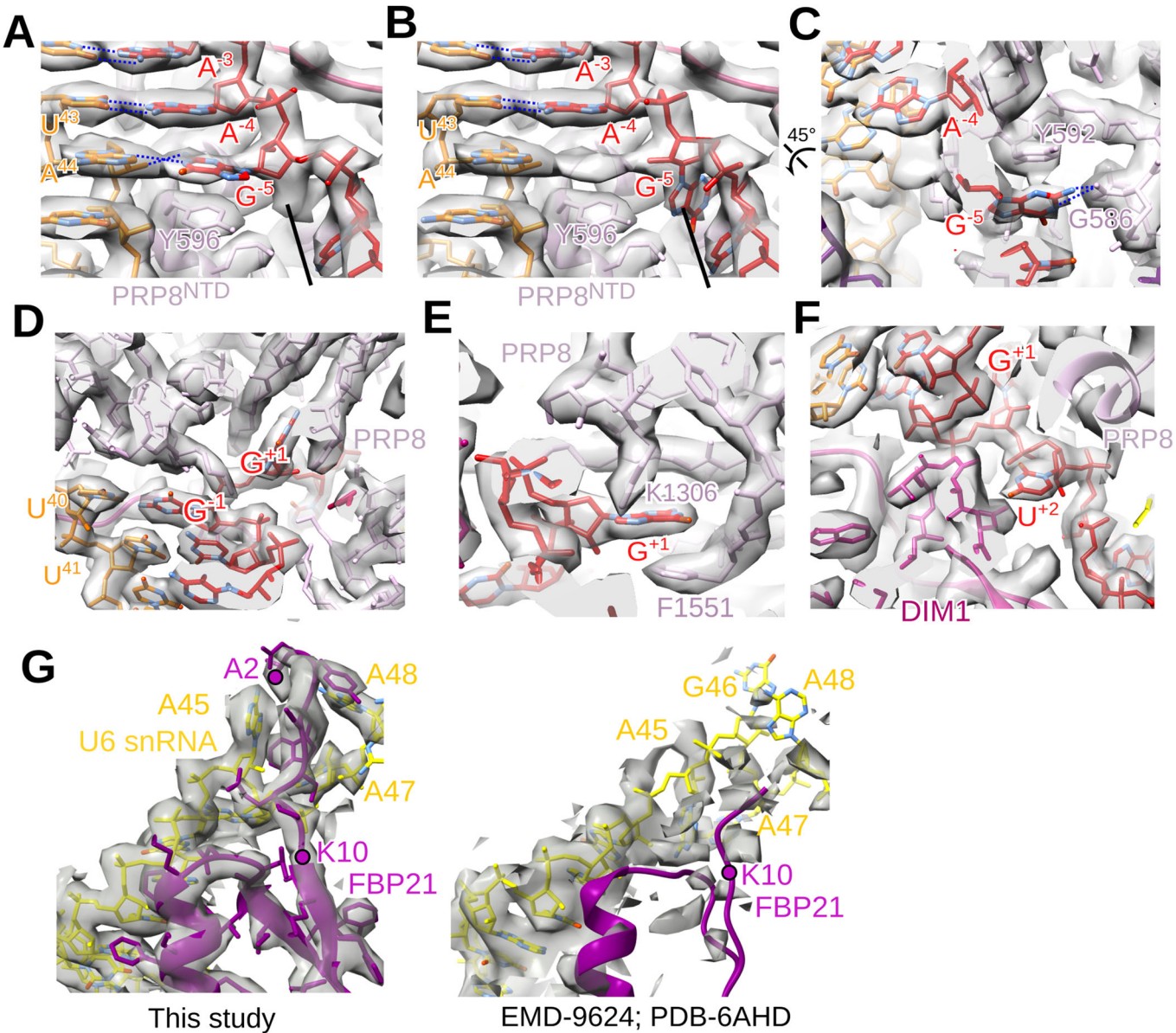

**Figure EV2. Molecular interactions with the 5′ss nucleotides in hB.**

(**A–C**) Two different conformations (panels **A** and **B**) of the base of G$^{-5}$ of the 5′ exon and their stabilization by neighboring residues of the PRP8 N-terminal domain (PRP8$^{NTD}$). The fit of U5 snRNA (orange) and 5′ exon nucleotides (red), where numbers indicate the nucleotide position relative to the 5′ss GU dinucleotide at the 5′end of the intron, to the EM density is shown. Dashed lines, hydrogen bonds. Panel (**C**) shows a different view of the conformation shown in panel (**B**). In one conformation, G$^{-5}$ base pairs with U5-A44 via two hydrogen bonds, one with the base of U5-A44, and the other with the preceding phosphate group. Moreover, the position of the G$^{-5}$ base is further stabilized by stacking interactions with Y592 of PRP8$^{NTD}$ (panel **A**). In the alternate conformation, the G$^{-5}$ base is flipped back by about 0.6 nm (panel **B**) and is stabilized by stacking interactions with Y592 and by hydrogen bonds with the protein main chain at G586 of PRP8$^{NTD}$ (panel **C**). The functional relevance of these alternative conformations, is not clear. However, after step 1 (i.e., in the human spliceosomal C complex), the base-paired conformation appears to be favored (Bertram et al, 2020), which would help to tether the cleaved 5′ exon to the spliceosome prior to exon ligation. (**D–F**) Fit of 5′ss nucleotides and neighboring RNAs and proteins to the hB EM density. (**G**) Fit of the N-terminus of FBP21, together with the U6 nts with which it interacts, into the hB EM density (left panel). Comparison with a previous hB complex model (PDB-GAHD) (Zhan et al, 2018), in which U6 snRNA nts 46–47 were incorrectly placed in the density occupied by the N-terminus of FBP21 (right panel). In the B complex from the yeast *S. cerevisiae*, the 5′ss and U6 ACAGA box region, as well as the 5′ exon and 5′ stem-loop of the U6 snRNA, appear to be organized somewhat differently compared to the hB complex (Plaschka et al, 2017; Bai et al, 2018).

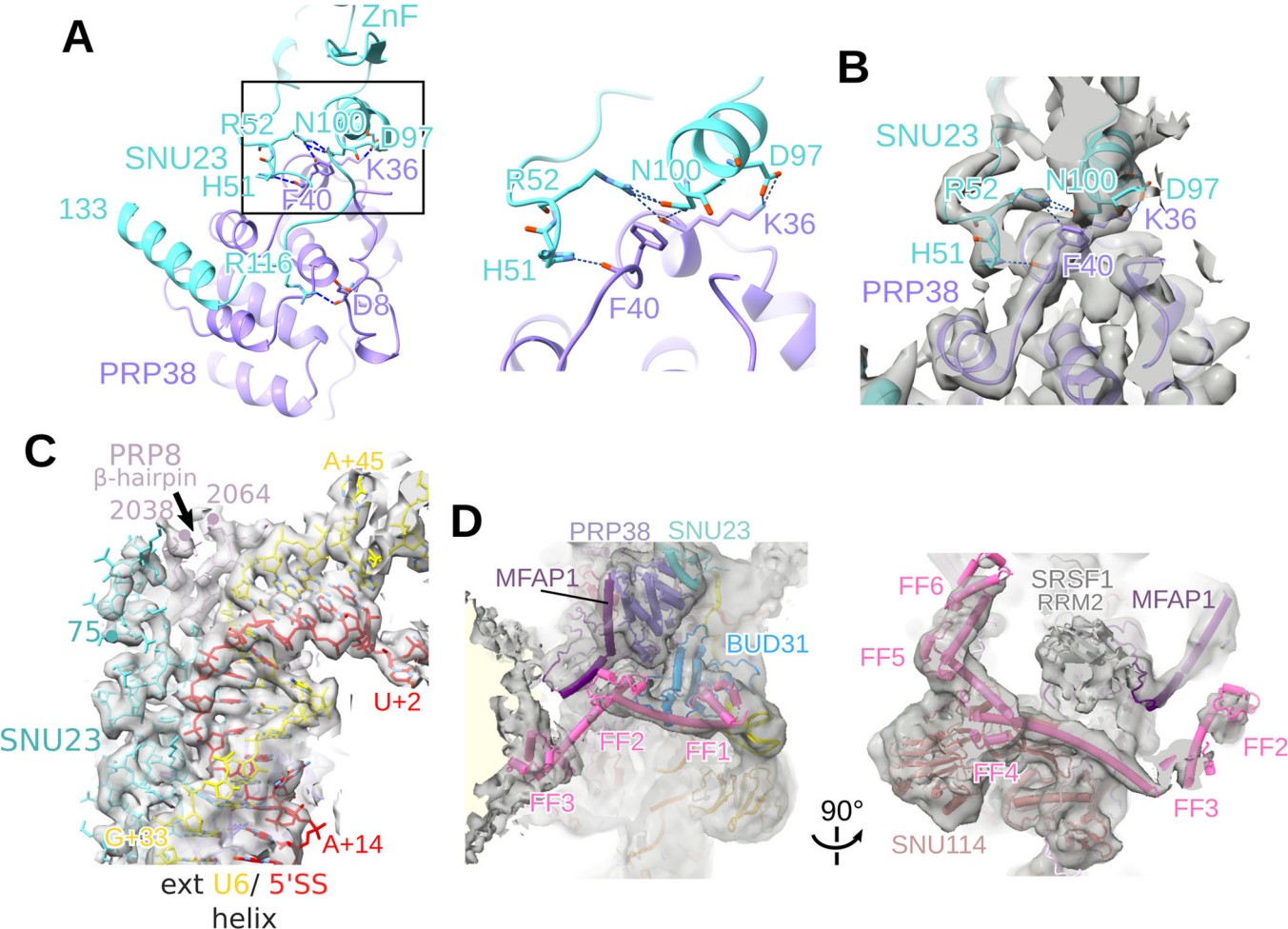

**Figure EV3. Interface of the SNU23 zinc finger and PRP38 N-terminal region, and localization of TCERG1 and SRSF1.**

(A) Interface between the SNU23 zinc finger (ZnF) and PRP38 N-terminal region. The boxed region is expanded in the right panel. (B) Fit of the SNU23 and PRP38 residues shown at the right in panel (A) into the hB EM density. (C) Fit of SNU23, the PRP8 RNase H-Jab1 linker and the U6/5′ss helix to the hB EM density. (D) Fit of the TCERG1 FF1-FF6 domains and SRSF1 RRM2 into the EM density of the hB monomer. Although the α-helix bridging FF2 and FF3 cannot be visualized in hB, fitting of the FF1-FF3 structure from pre-B$^{act-1}$ into our B complex indicates that FF3 is located in a small globular density that directly contacts the lower globule 2.

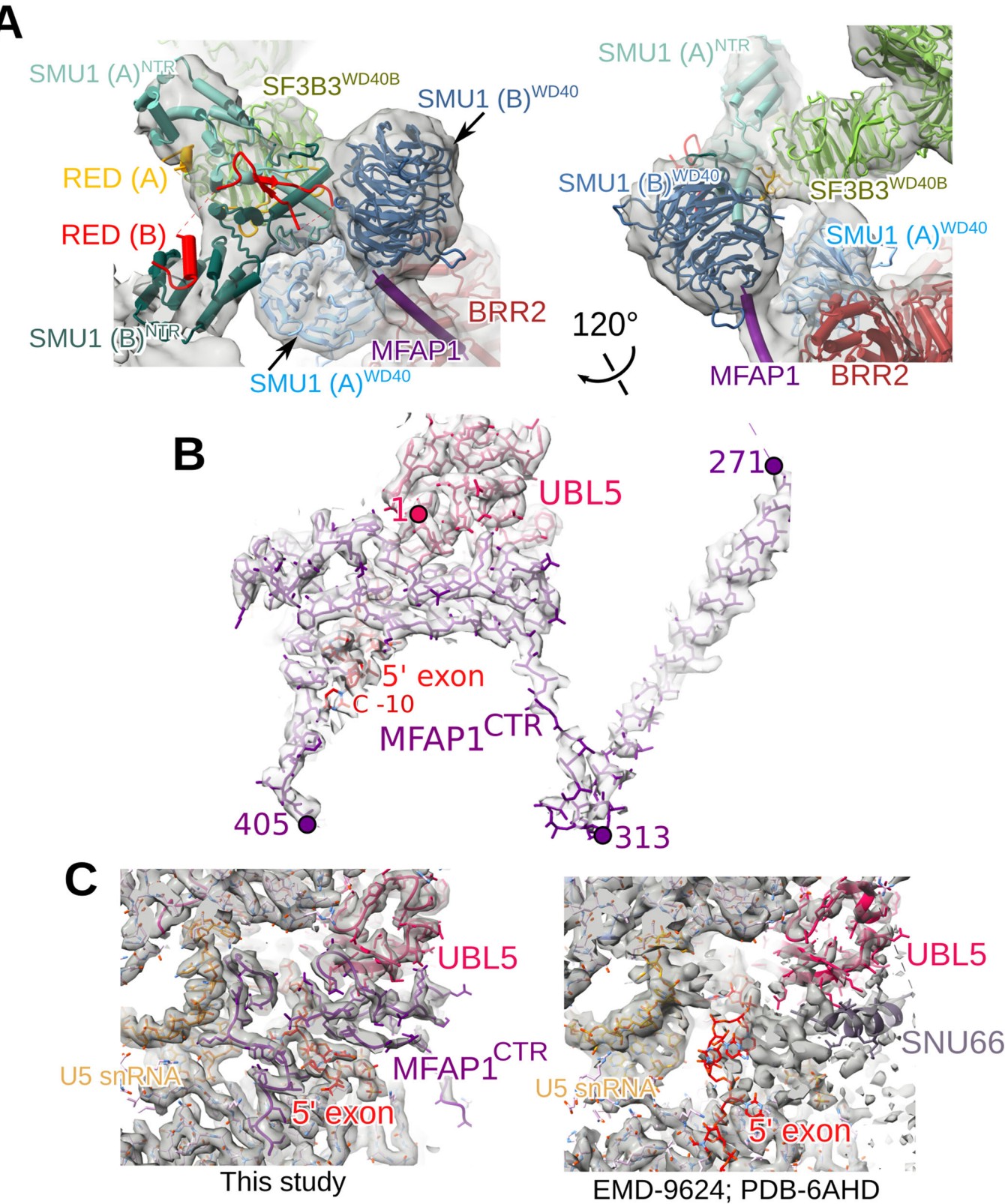

**Figure EV4.   A heterotetrameric SMU1-RED complex is present in the human B complex.**

(A) Fit of the SMU1 N-terminal region (NTR) and the WD40 domain of each SMU1 subunit (denoted **A** or **B**) of the SMU1 dimer, plus 2 copies of RED, into the hB EM density. The orientation of the individual NTR domains (**A** or **B**) cannot be determined unambiguously. That the B-specific proteins SMU1 and RED interact with each other was initially shown by co-precipitation experiments (Chung et al, 2009) and luciferase complementation assays (Fournier et al, 2014). The crystal structure of a minimal SMU1-RED complex from C. elegans and human, comprised of the SMU1 N-terminal region and a middle region of RED, showed that SMU1 dimerizes via its LiSH motif and a C-terminal α-helix, and that the dimerization module binds two copies of RED (Ashraf et al, 2019; Ulrich et al, 2016a). The human structure suggests that the SMU1 N-terminal region must first form a dimer before it can interact with two copies of the middle domain of RED (Ashraf et al, 2019). (B) Fit of MFAP1 helix 217–313 and its C-terminal region (CTR), as well as UBL5 and the 3′ end of the 5′ exon, to the hB EM density. (C) Fit of amino acids in the MFAP1 C-terminal region to the hB EM density. Comparison with a previous hB complex model (PDB-6AHD) (Zhan et al, 2018), in which an α-helix of SNU66 was incorrectly placed in the density occupied by the MFAP1 C-terminal region.

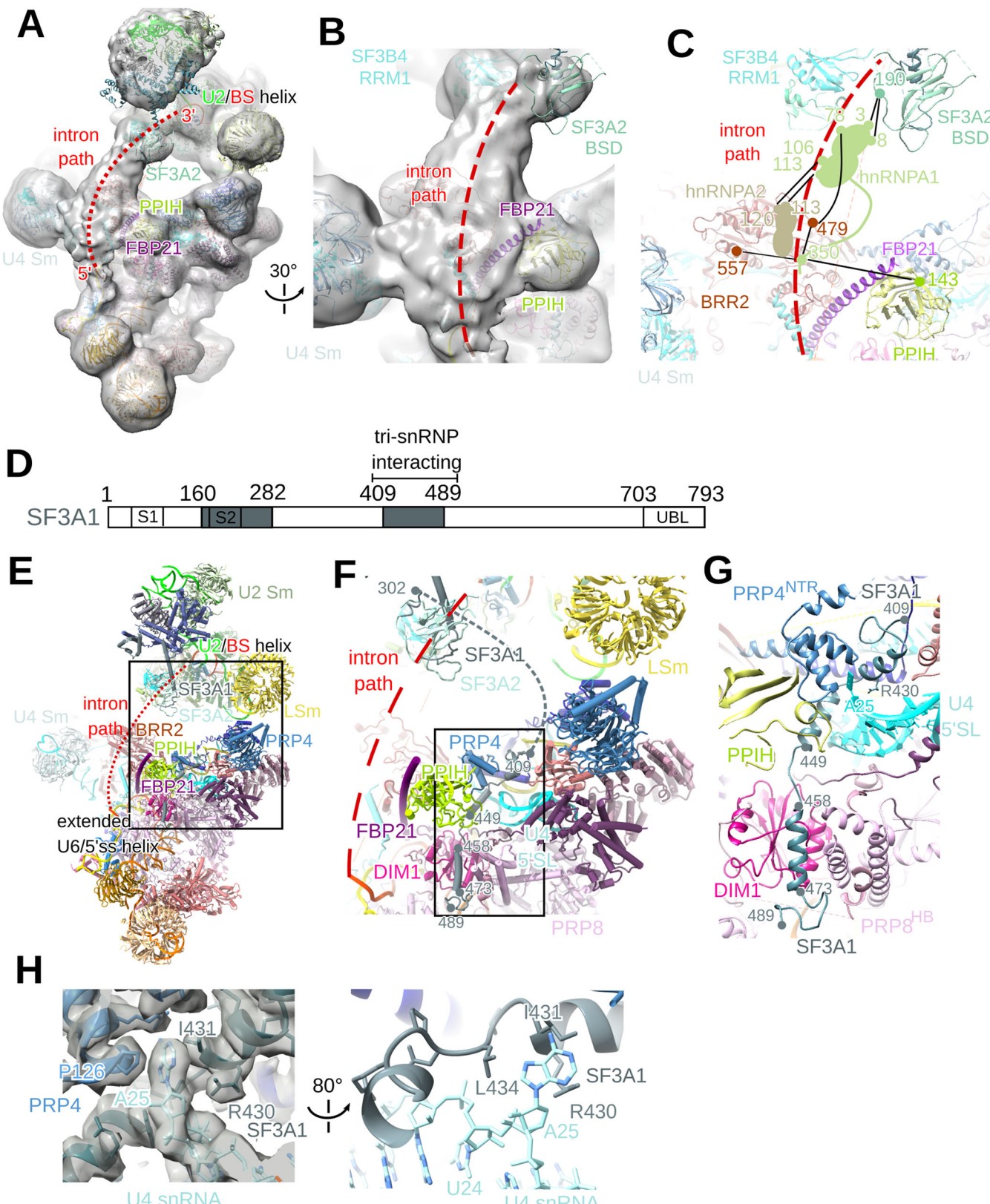

**Figure EV5.   Location of intron nucleotides upstream of the branch site and a C-terminal region of SF3A1 in hB.**

(A) Proposed path (dashed line) of the intron between the U2/BS helix and U6/5′ss helix. (B, C) Intron nucleotides directly upstream of the BS are sandwiched between SF3B4 RRM1 and the β-sandwich domain (BSD) of SF3A2, and nucleotides further upstream are likely contacted by hnRNP proteins. Panel (B) shows the EM density that appears to accommodate the intron, as well as adjacent protein domains. Panel (C) shows the proposed position of hnRNPA1 and A2, based on protein crosslinks with neighboring B complex proteins. Intermolecular crosslinks are indicated by a line, with the position of the crosslinked residues indicated. (D) Domain organization of SF3A1. The modeled regions are indicated as colored boxes. S1 and S2, SURP domains 1 and 2; UBL, ubiquitin-like domain. (E) Overview of the hB complex molecular architecture. The boxed region is expanded in panel (F). (F, G). SF3A1 aa 409–449 interact with U4 SL1, PRP4 and PPIH, while SF3A1 aa 454–489, which form an α-helix, are located between DIM1 and the PRP8 helical bundle (PRP8HB). The region boxed in panel (F) is expanded in panel (G). (H) Fit of indicated U4 nucleotides, and amino acids of PRP4 and SF3A1 into the hB EM density. The base of A25 interacts with several hydrophobic protein side chains (i.e., I431, L434 of SF3A1 and P126 of PRP4), while R430 of SF3A1 contacts the backbone of U4-A25.

