## [Peer Review File · The EMBO Journal]

Cryo-EM analyses of dimerized spliceosomes provide new insights into the functions of B complex proteins

Reinhard Luehrmann, Zhenwei Zhang, Vinay Kumar, Olexandr Dybkov, Cindy Will, Henning Urlaub, and Holger Stark

Corresponding author(s): Reinhard Luehrmann (Reinhard.Luehrmann@mpinat.mpg.de) , Holger Stark (hstark1@mpinat.mpg.de)

Review Timeline:

Submission Date:	27th Sep 23
Editorial Decision:	2nd Nov 23
Revision Received:	15th Dec 23
Editorial Decision:	18th Jan 24
Revision Received:	25th Jan 24
Accepted:	26th Jan 24

Editor: William Teale

Transaction Report:

Dear Prof. Luehrmann,

Thank you again for the submission of your manuscript entitled "New insights into the functions of B complex proteins revealed by cryo-EM of dimerized spliceosomes" and for your patience during the review process. We have now received reports from three referees, which I copy below.

As you can see from their comments, while the referees make some suggestions on how you might consider strengthening the manuscript, all are broadly supportive of your experimental approach.

Therefore, based on the overall interest expressed in the reports, I would like to invite you to address the comments of all referees in a revised version of the manuscript. I should add that it is The EMBO Journal policy to allow only a single major round of revision and that it is therefore important to resolve the main concerns at this stage. I believe the concerns of the referees are reasonable and addressable, but please contact me if you have any questions, need further input on the referee comments or if you anticipate any problems in addressing any of their points. Please, follow the instructions below when preparing your manuscript for resubmission.

I would also like to point out that as a matter of policy, competing manuscripts published during this period will not be taken into consideration in our assessment of the novelty presented by your study ("scooping" protection). We have extended this 'scooping protection policy' beyond the usual 3 month revision timeline to cover the period required for a full revision to address the essential experimental issues. Please contact me if you see a paper with related content published elsewhere to discuss the appropriate course of action.

Again, please contact me at any time during revision if you need any help or have further questions.

Thank you very much again for the opportunity to consider your work for publication. I look forward to your revision.

Best regards,

William

William Teale, Ph.D.
Editor
The EMBO Journal

When submitting your revised manuscript, please carefully review the instructions below and include the following items:

- 1) a .docx formatted version of the manuscript text (including legends for main figures, EV figures and tables). Please make sure that the changes are highlighted to be clearly visible.
- 2) individual production quality figure files as .eps, .tif, .jpg (one file per figure).
- 3) a .docx formatted letter INCLUDING the reviewers' reports and your detailed point-by-point response to their comments. As part of the EMBO Press transparent editorial process, the point-by-point response is part of the Review Process File (RPF), which will be published alongside your paper.
- 4) a complete author checklist, which you can download from our author guidelines ([https://wol-prod-cdn.literatumonline.com/pb-assets/embo-site/Author Checklist%20-%20EMBO%20J-1561436015657.xlsx](https://wol-prod-cdn.literatumonline.com/pb-assets/embo-site/Author%20Checklist%20-%20EMBO%20J-1561436015657.xlsx)). Please insert information in the checklist that is also reflected in the manuscript. The completed author checklist will also be part of the RPF.
- 5) Please note that all corresponding authors are required to supply an ORCID ID for their name upon submission of a revised manuscript.
- 6) We require a 'Data Availability' section after the Materials and Methods. Before submitting your revision, primary datasets produced in this study need to be deposited in an appropriate public database, and the accession numbers and database listed under 'Data Availability'. Please remember to provide a reviewer password if the datasets are not yet public (see <https://www.embopress.org/page/journal/14602075/authorguide#datadeposition>). If no data deposition in external databases is

needed for this paper, please then state in this section: This study includes no data deposited in external repositories. Note that the Data Availability Section is restricted to new primary data that are part of this study.

Note - All links should resolve to a page where the data can be accessed.

8) For data quantification: please specify the name of the statistical test used to generate error bars and P values, the number (n) of independent experiments (specify technical or biological replicates) underlying each data point and the test used to calculate p-values in each figure legend. The figure legends should contain a basic description of n, P and the test applied. Graphs must include a description of the bars and the error bars (s.d., s.e.m.).

9) We would also encourage you to include the source data for figure panels that show essential data. Numerical data can be provided as individual .xls or .csv files (including a tab describing the data). For 'blots' or microscopy, uncropped images should be submitted (using a zip archive or a single pdf per main figure if multiple images need to be supplied for one panel). Additional information on source data and instruction on how to label the files are available at .

10) We replaced Supplementary Information with Expanded View (EV) Figures and Tables that are collapsible/expandable online (see examples in <https://www.embopress.org/doi/10.15252/embj.201695874>). A maximum of 5 EV Figures can be typeset. EV Figures should be cited as 'Figure EV1, Figure EV2" etc. in the text and their respective legends should be included in the main text after the legends of regular figures.

12) Our journal encourages inclusion of *data citations in the reference list* to directly cite datasets that were re-used and obtained from public databases. Data citations in the article text are distinct from normal bibliographical citations and should directly link to the database records from which the data can be accessed. In the main text, data citations are formatted as follows: "Data ref: Smith et al, 2001" or "Data ref: NCBI Sequence Read Archive PRJNA342805, 2017". In the Reference list, data citations must be labeled with "[DATASET]". A data reference must provide the database name, accession number/identifiers and a resolvable link to the landing page from which the data can be accessed at the end of the reference. Further instructions are available at .

Further instructions for preparing your revised manuscript:

We realize that it is difficult to revise to a specific deadline. In the interest of protecting the conceptual advance provided by the work, we recommend a revision within 3 months (31st Jan 2024). Please discuss the revision progress ahead of this time with the editor if you require more time to complete the revisions.

Referee #1:

In this work, Zhang and coworkers obtain a new spliceosome structure composed of dimers of B complexes connected by globular domains at the top and bottom. The role of the globular domains is ambiguous; however, the B complex structures themselves are highly important for the field since they provide significantly more insight into the structure of this complex and how splice sites are being recognized and ultimately transferred to the catalytic site. The manuscript does an excellent job of describing the structures in detail as well as the significance of the major new interactions that are observed. Moreover, the cryo-EM appears to be top-notch and supported by mass spec data. I think this is a very important paper in the field that both clarifies existing structural data, provides new structural insights, and allows for lots of hypotheses to be generated and tested for how splice site recognition occurs and can be modulated. Experimentally, I don't think there is anything more to add to this work. However, I do suggest some edits or additions as indicated below. To me the main concern about this manuscript is that it is very dense, especially for outsiders to the field. While the results are significant, I think perhaps a "less is more" approach and putting key results in context of a larger picture accessible to a wide audience could increase its impact. Nonetheless, this is terrific structural biology and an important advance for the field.

Major Points:

1. I didn't see a reference to the work of Ruth Sterling here and her supraspliceosome model. Is there a connection? Are the data here consistent with such supraspliceosomes?
2. In general I found the writing of the manuscript excellent but often went into the "weeds" of splicing and lost sight of the big picture. I think taking a step back and asking about how to make the manuscript more accessible to those new to or outside of the splicing field would greatly improve its impact.
3. The interactions of the 5'SS are very interesting and a highlight of its work. This is reminiscent of a model (I think from a prior Luhrmann lab paper?) of repeated rounds of recognition and handover of the splice sites between splicing factors. It would perhaps be interesting to highlight these findings in context of this and this could help provide a frame of reference for those outside of the field.
4. The authors state that when using a MINX exon RNA substrate that lacks the 5' exon, the lower globular density is unobserved. In my view this is pretty important evidence in support of the author's assignment of the globular densities. This should be included in the main text or, at minimum, a supplemental figure.

5. I found the discussion on page 18 concerning "clusters", nuclear speckles, and phase separation to be misleading. The authors should think about the size of the particles observed here (perhaps in volume) vs. the volume of a nuclear speckle inside a cell. My gut instinct is that there is a large mismatch in the molecular sizes of the species being compared. This is so speculative and potentially misleading that it should be removed from the discussion.

Minor Points and Suggestions:

1. I found the use of AlphaFold compelling. It would be interesting to include a supplemental figure showing the direct alpha fold predictions vs. what was finally modeled in the structure based on the observed cryo-EM density. Was Alpha Fold ever totally "wrong"? If so, how?
2. I think having a figure showing the previously published B complex structure (perhaps all in grey) vs. everything new in this structure (perhaps in grey with the new stuff in another color) would help highlight for the reader what exactly is new here.
3. It would be very useful to compare the B complex proteins here and their interactions with the 5'SS with the B complex proteins observed in yeast spliceosome structures, including those purified under limiting ATP. I believe that in the yeast structures, the 5' exon is in a very different location as is the 5' stem loop of U6. I think this needs to be commented on at some point from a structural perspective.

Referee #2:

The manuscript "New insights into the functions of B complex proteins revealed by cryo-EM of dimerized spliceosomes" by Zhang, et al. describes an improved structural model of the human spliceosome stalled before catalytic activation. The improvements are derived from increased cryo-EM map resolution derived from increased structural rigidity that results from B complex dimerization, although the biological relevance of the dimerization is not known. The work builds on previously published models of B complex by increasing the level of detail of important region of the models, extending previously modeled proteins, adding a few new proteins, and correcting previously misinterpreted density. The authors strive to provide context for the newly modeled densities, although the functional relevance of many interactions remain speculative without further experimentation. Overall, the improved model represents an important advance in visualization of key interactions in B complex, and it will be welcomed by researchers aiming to generate and interpret genetic and biochemical data related to the specific roles of B complex protein and RNA interactions. My major critique is that the manuscript is very challenging to digest given the alphabet soup of spliceosome complexes, the large number of individual proteins with names that vary in the literature and often have complex domain structures. The authors work to overcome these issues with figures that include diagrams of protein domains and close-up views of key interactions. However, the findings could be better communicated to researchers not fully steeped in the complexity of the spliceosome by addressing the following points:

1. More cartoon models, like the one for SF3B6 positioning in Figure 7D, would be very helpful for illustrating the structural changes that accompany moving into and out of B complex, especially as they pertain to the newly modeled proteins and interactions described in the manuscript.
2. An illustration of what has changed between this new structure and previous models would also help readers understand what is new in this work.
3. In my opinion, the text would flow better for most readers by using full names of protein domains rather than using abbreviations for individual protein domains (ex. PRP8 RNase H domain vs. PRP8RH).
4. It would be helpful to include gene names when they differ from the protein name (ex. ZMAT2 for SNU23), at least in Figure 1.
5. What is the relevance of the two conformations of the G-5 nucleotide? I could not decipher the evidence for the difference from the figure. Could this be due to averaging of the two protomers?

Referee #3:

SUMMARY: This is a significant advance over prior, low-resolution cryo-EM structures of the human B complex spliceosome. By using ATP γ S, in vitro assembled spliceosomes stall at the B complex stage that dimerize. By masking one subunit, the authors were able to improve the local resolutions, spanning 2.9-3.5 angstrom resolution. The resulting series of structures were then modeled from known protein structures, homology modeling, and AlphaFold modeling. Crosslinked mass spectrometry was used to validate the models. From the resulting model, the authors were able to propose novel hypotheses about how the 5' ss is selected to prime it for catalysis in the subsequent complexes; explain how the B-complex dimerizes; position the 5' exon before

it is liberated from the intron; and allow modeling of protein factors that were previously not resolved.

MAJOR CONCERNS: The conclusions seem warranted given the improved resolution of this series of structures. There are no major concerns from this reviewer.

MINOR CONCERNS: The manuscript is a bit of an alphabet soup that might be technically difficult to parse for a general reader. Additional information for table S1 that briefly generalizes their role in B complex might help the reader follow along.

Is there a reason for the dramatic downward spike in the BRR2 FSC (Figure S1F)?

The results and discussion are merged. Is this appropriate for EMBO? This reviewer was under the impression that this composition was rather out of date.

In summary, this is a technically-sound structural study of B-complex spliceosomes that is suitable for publication in EMBO J.

MAX-PLANCK-INSTITUT für Multidisziplinäre Naturwissenschaften
KARL-FRIEDRICH-BONHOEFFER-INSTITUT

MPI für für Multidisziplinäre Naturwissenschaften, 37077 Göttingen

William Teale
Editor, EMBO J

Prof. Dr. Reinhard Lührmann
Zelluläre Biochemie

Am Faßberg 11 37077 Göttingen
Telefon: (0551) 201-1407
Telefax: (0551) 201-1197
E-Mail: reinhard.luehrmann@mpinat.mpg

December 14, 2023

Dear Dr. Teale,

We were very pleased that our manuscript entitled “**New insights into the functions of B complex proteins revealed by cryo-EM of dimerized spliceosomes**” was well received by the Reviewers and to hear that a revised version of our manuscript may potentially be suitable for publication in EMBO J. In the revised version we have made several changes/improvements as requested by the Reviewers and include a point-by-point response to their comments as outlined below:

Referee #1:

In this work, Zhang and coworkers obtain a new spliceosome structure composed of dimers of B complexes connected by globular domains at the top and bottom. The role of the globular domains is ambiguous; however, the B complex structures themselves are highly important for the field since they provide significantly more insight into the structure of this complex and how splice sites are being recognized and ultimately transferred to the catalytic site. The manuscript does an excellent job of describing the structures in detail as well as the significance of the major new interactions that are observed. Moreover, the cryo-EM appears to be top-notch and supported by mass spec data. I think this is a very important paper in the field that both clarifies existing structural data, provides new structural insights, and allows for lots of hypotheses to be generated and tested for how splice site recognition occurs and can be modulated. Experimentally, I don't think there is anything more to add to this work. However, I do suggest some edits or additions as indicated below. To me the main concern about this manuscript is that it is very dense, especially for outsiders to the field. While the results are significant, I think perhaps a "less is more" approach and putting key results in context of a larger picture accessible to a wide audience could increase its impact. Nonetheless, this is terrific structural biology and an important advance for the field.

Major Points:

1. I didn't see a reference to the work of Ruth Sterling here and her supraspliceosome model. Is there a connection? Are the data here consistent with such supraspliceosomes?

Supraspliceosomes are large (ca 200 S) endogeneous complexes, isolated from cell nuclei, that appear to consist of four spliceosomal subunits that form on a single pre-mRNA (Sebbag-Sznajder et al., 2020, Front. Genet.). The spliceosomal B complex dimers that we describe here are formed from two spliceosomes that are each assembled in vitro on a separate pre-mRNA substrate, and they sediment at a much lower S value (Agafonov et al., 2016, RNA). They are thus clearly different from the supraspliceosomes described by Ruth Sperling's group, and for this reason we did not refer to their work.

2. In general I found the writing of the manuscript excellent but often went into the "weeds" of splicing and lost sight of the big picture. I think taking a step back and asking about how to make the manuscript more accessible to those new to or outside of the splicing field would greatly improve its impact.

The spliceosome is a highly complex molecular machine. Our improved resolution of the human spliceosomal B complex elucidates many new, functionally important details of the splicing machinery and, thus, it is clearly necessary that we describe these details (and go into the "weeds"). With the goal of making the manuscript more accessible to those new to or outside the splicing field, we have added several sentences to the revised version pointing out how our results illustrate more general principles of how the spliceosome and splicing process works. We have also added a table summarizing the known roles of most of the B complex proteins described in the manuscript (Appendix Table S1) and an additional figure summarizing the major structural and compositional changes during B complex formation (Appendix Figure S1).

3. The interactions of the 5'SS are very interesting and a highlight of its work. This is reminiscent of a model (I think from a prior Luhrmann lab paper?) of repeated rounds of recognition and handover of the splice sites between splicing factors. It would perhaps be interesting to highlight these findings in context of this and this could help provide a frame of reference for those outside of the field.

We now include several sentences on p. 9 pointing out that the 5'ss GU dinucleotides are subsequently handed off from DIM1 and PRP8 to RNF113A and SF3A2 during the transformation of hB to hBact, and that sequestration by different proteins before the first catalytic step of splicing prevents a premature nucleophilic attack on the 5'ss.

4. The authors state that when using a MINX exon RNA substrate that lacks the 5' exon, the lower globular density is unobserved. In my view this is pretty important evidence in support of the author's assignment of the globular densities. This should be included in the main text or, at minimum, a supplemental figure.

Cryo-EM of spliceosomes formed on a MINX exon RNA substrate is summarized in a second manuscript that is currently under peer review at a different journal. As we stated that the data in that manuscript are not under consideration elsewhere, I am afraid that we are currently not at liberty to show these data prior to their publication. If it is not possible to simply refer to these data as "Zhang et al, unpublished" then we could also delete this sentence. Even in the absence of this additional information, the data presented in this manuscript clearly support our conclusion that the upper globular density element that connects the two protomers in the B dimer is comprised of the two 3' exons and associated binding proteins, whereas the lower globular density element is comprised of the two 5' exons and associated binding proteins.

5. I found the discussion on page 18 concerning "clusters", nuclear speckles, and phase separation to be misleading. The authors should think about the size of the particles observed here (perhaps in volume) vs. the volume of a nuclear speckle inside a cell. My gut instinct is that there is a large mismatch in the molecular sizes of the species being compared. This is so speculative and potentially misleading that it should be removed from the discussion.

We did not mean to imply that our poorly-defined globules are nuclear speckles, rather that they may be a kind of biomolecular condensate. To avoid any confusion, we have modified the text and no longer refer to nuclear speckles. However, we find it important to address what our diffuse globules might represent and thus we still speculate that they might potentially be biomolecular condensates.

Minor Points and Suggestions:

1. I found the use of AlphaFold compelling. It would be interesting to include a supplemental figure showing the direct alpha fold predictions vs. what was finally modeled in the structure based on the observed cryo-EM density. Was Alpha Fold ever totally "wrong"? If so, how?

We find it beyond the scope of this manuscript to describe in detail how well certain regions modeled by AlphaFold (AF) actually fit into our EM density. There are published papers that assess in detail the quality of AF2 predictions (e.g., Akdel *et al.*, 2022, NSMB). AF2 predictions are very accurate for globular domains that do not have much structural variation. For domains that can change their conformation (e.g., the HAT domain), AF can accurately predict the local folding, but not the overall conformation, as it lacks the neighboring protein context. The same holds true for less stable regions. AF2 can detect loops, hairpins, and short α -helices very accurately, but the relative positions between these local motifs always needs to be adjusted accordingly. Those protein regions for which AF predictions were used for modelling of our hB complex are summarized in Appendix Table S4. From the latter it is clear whether or not the predicted AF structures fit into our EM density without (i.e., designated "docked") or with (i.e., "docked and adjusted") the need for further adjustments.

2. I think having a figure showing the previously published B complex structure (perhaps all in grey) vs. everything new in this structure (perhaps in grey with the new stuff in another color) would help highlight for the reader what exactly is new here.

Instead of showing such a figure, in which it is hard to discern exactly which protein/RNA regions are new or improved, we now include a detailed Table (Appendix Table S5) that lists all of the improvements and newly-modeled protein and RNA regions compared to the previously-published hB complex structures.

3. It would be very useful to compare the B complex proteins here and their interactions with the 5'SS with the B complex proteins observed in yeast spliceosome structures, including those purified under limiting ATP. I believe that in the yeast structures, the 5' exon is in a very different location as is the 5' stem loop of U6. I think this needs to be commented on at some point from a structural perspective.

As our manuscript focusses on the human spliceosome, and differences between the human and yeast B complex have been discussed elsewhere (reviewed in Wilkinson *et al.*, 2020, *Ann. Rev.*; Wan *et al.*, 2020, *Ann. Rev.*), we would argue that a lengthy, detailed description about these differences is not warranted. Furthermore, there is some disagreement concerning the RNP architecture at/or near the 5'ss in the yeast spliceosome (Plaschka *et al.*, 2017, *Nature*; Bai *et al.*, 2018, *Science*). Thus, we have chosen not to make any detailed comparisons with the situation in the human B complex. Rather we now state in the legend to Figure EV2 more generally that "In the B complex from the yeast *S. cerevisiae*, the 5'ss and U6 ACAGA box region, as well as the 5' exon and 5' SL of the U6 snRNA appear to be organized somewhat differently compared to the hB complex (Plaschka *et al.*, 2017, Bai *et al.*, 2018)."

Referee #2:

The manuscript "New insights into the functions of B complex proteins revealed by cryo-EM of dimerized spliceosomes" by Zhang, et al. describes an improved structural model of the human spliceosome stalled before catalytic activation. The improvements are derived from increased cryo-EM map resolution derived from increased structural rigidity that results from B complex dimerization, although the biological relevance of the dimerization is not known. The work builds on previously published models of B complex by increasing the level of detail of important region of the models, extending previously modeled proteins, adding a few new proteins, and correcting previously misinterpreted density. The authors strive to provide context for the newly modeled densities, although the functional relevance of many interactions remain speculative without further experimentation. Overall, the improved model represents an important advance in visualization of key

interactions in B complex, and it will be welcomed by researchers aiming to generate and interpret genetic and biochemical data related to the specific roles of B complex protein and RNA interactions. My major critique is that the manuscript is very challenging to digest given the alphabet soup of spliceosome complexes, the large number of individual proteins with names that vary in the literature and often have complex domain structures. The authors work to overcome these issues with figures that include diagrams of protein domains and close-up views of key interactions. However, the findings could be better communicated to researchers not fully steeped in the complexity of the spliceosome by addressing the following points:

1. More cartoon models, like the one for SF3B6 positioning in Figure 7D, would be very helpful for illustrating the structural changes that accompany moving into and out of B complex, especially as they pertain to the newly modeled proteins and interactions described in the manuscript.

We now include a schematic of the main structural changes occurring during the pre-B to B complex transition in Appendix Fig. S1. The structural and compositional changes accompanying the B to Bact transition are well documented in Townsend et al. 2020, and thus we did not include them here. As essentially all of the main figures show protein-protein or protein-RNA interactions within the hB complex that are in most cases depicted as ribbon diagrams or space filling models, we do not find it necessary to additionally show them as cartoon models in the main figures. A cartoon overview of the locations/interactions of several of the B-specific proteins is shown in Appendix Figure S1D.

2. An illustration of what has changed between this new structure and previous models would also help readers understand what is new in this work.

See our response above to Referee 1's minor comment #2.

3. In my opinion, the text would flow better for most readers by using full names of protein domains rather than using abbreviations for individual protein domains (ex. PRP8 RNase H domain vs. PRP8RH).

We have deleted most of these abbreviations, except in the figures where this is not feasible, and now use the full names of most of the protein domains.

4. It would be helpful to include gene names when they differ from the protein name (ex. ZMAT2 for SNU23), at least in Figure 1.

All gene names are shown in Appendix Table S2. We have added a sentence to the legend to Figure 1 referring the reader to Appendix Table S2 for gene names.

5. What is the relevance of the two conformations of the G-5 nucleotide? I could not decipher the evidence for the difference from the figure. Could this be due to averaging of the two protomers?

While we cannot exclude that one protomer adopts one conformation, while the other adopts the other conformation, it is more likely that within both protomers G5 can naturally adopt two different conformations that are stabilized by two different tyrosines. Concerning the relevance of the two conformations, we state in the legend to EV Fig. 2 that "The functional relevance of these alternative conformations, is not clear. However, after step 1 (i.e., in the human spliceosomal C complex), the base-paired conformation appears to be favored (Bertram *et al.* 2020), which would help to tether the cleaved 5' exon to the spliceosome prior to exon ligation."

Referee #3:

SUMMARY: This is a significant advance over prior, low-resolution cryo-EM structures of the human B complex spliceosome. By using ATP γ S, in vitro assembled spliceosomes stall at the B complex stage that dimerize. By masking one subunit, the authors were able to improve the local resolutions, spanning 2.9-3.5 angstrom resolution. The resulting series of structures were then modeled from known protein structures, homology modeling, and AlphaFold modeling. Crosslinked mass spectrometry was used to validate the models. From the resulting model, the authors were able to propose novel hypotheses about how the 5' ss is selected to prime it for catalysis in the subsequent complexes; explain how the B-complex dimerizes; position the 5' exon before it is liberated from the intron; and allow modeling of protein factors that were previously not resolved.

MAJOR CONCERNS: The conclusions seem warranted given the improved resolution of this series of structures. There are no major concerns from this reviewer.

MINOR CONCERNS:

The manuscript is a bit of an alphabet soup that might be technically difficult to parse for a general reader. Additional information for table S1 that briefly generalizes their role in B complex might help the reader follow along.

We now include a separate Table (Appendix Table S1) summarizing the known roles of a subset of the B complex proteins.

Is there a reason for the dramatic downward spike in the BRR2 FSC (Figure S1F)?

We do not have a mathematical explanation for the downward spike. But such a spike is not unusual for masked refinement of a smaller region of a large complex, and is also observed in a paper describing the cryo-EM structure of the human pre-B complex (Charenton *et al.*, 2019, Fig. S4D).

The results and discussion are merged. Is this appropriate for EMBO? This reviewer was under the impression that this composition was rather out of date.

EMBO J allows the Results and Discussion to be merged.

In summary, this is a technically-sound structural study of B-complex spliceosomes that is suitable for publication in EMBO J.

We thank the referees for their constructive criticisms. We hope our manuscript is now suitable for publication in EMBO J.

Sincerely,

Reinhard Lührmann
Holger Stark

Dear Reinhard,

Thank you for submitting the revised version of your manuscript, which addresses the concerns of the referees. This revised version has now been re-reviewed; I attach the second referee reports to the bottom of this mail. As you will see, you have addressed the referees' concerns to their satisfaction. Reviewer 1 makes some final constructive suggestions which I would like you to consider carefully. Before I can finally accept the manuscript, there are some remaining editorial points which need to be addressed. In this regard, would you please:

- rename the 'Conflict of Interest' section the 'Disclosure Statement and Competing Interests' statement,
- remove the author credit section from the manuscript file,
- rename the callout for Table S4 as 'Appendix Table S4',
- include dataset EV legends as a separate tab in the relevant Excel files,
- include page numbers missing in the table of contents in Appendix 1,
- provide specific URLs for Electron Microscopy Data Bank (EMDB) accession codes EMD19063, EMD-18529, EMD-18225, & EMD-19062 and Protein Data Bank (PDB) entries with the identifiers 8Q7N & 8QO9 datasets in the data availability statement, and
- ensure that all public data sets are made fully available upon acceptance of the manuscript.

I look forward to receiving these changes. EMBO Press is an editorially independent publishing platform for the development of EMBO scientific publications.

Best wishes,

William

William Teale, PhD
Editor
The EMBO Journal
w.teale@embojournal.org

We realize that it is difficult to revise to a specific deadline. In the interest of protecting the conceptual advance provided by the work, we recommend a revision within 3 months (17th Apr 2024). Please discuss the revision progress ahead of this time with

the editor if you require more time to complete the revisions.

Referee #1:

The authors have responded to the referee comments and I think the manuscript is ready for publication. I just have a few minor things that the authors may want to consider at their own discretion.

1. I still find the possible connection to Ruth Sperling's work fascinating. The authors should note that given the symmetry of their structure it should easily be able to be expanded to a 4 spliceosome complex (but I think a 3 spliceosome complex would not be symmetry permitted) and that intramolecular formation of such a complex on the same (large) transcript should be favored over formation of it intermolecularly between different RNAs. In any case, I think it is worth a reference.

2. I don't like the idea of presenting data in favor of a conclusion that can't be shown or evaluated by reviewers. I would urge the authors to remove the sentence referring to the unpublished work on the MINX exon RNA substrate that lacked the 5' exon. I think the data is sufficient for showing the correct assignment.

3. I still think it is misleading to propose that a spliceosome dimer may be phase separated in the absence of any supporting data. I suggest the following edit: "It is thus tempting to speculate that the diffuse globules that we observe in our hB dimers IN VITRO may CONTRIBUTE to bimolecular condensates containing exon RNA and, among others, SR proteins OBSERVED IN VIVO."

4. I found Table S5 to be quite useful. It is still unclear to me why this information can't be represented in a (potentially quite useful) figure or video. Can the authors define what docking means for non structural biologists? In addition, a few things were "modeled for the first time...". Does this mean that they were previously docked and not modeled or were they "docked and modeled for the first time"?

5. The cartoon figures in Appendix Figure S1 are so useful that they should be included in the main text figures. This would greatly facilitate the readability of the manuscript as well as increase the impact of the manuscript.

Referee #2:

The authors have addressed my concerns. I appreciate the additional information in the appendices, which should help guide readers in navigating these important spliceosome models.

Referee #3:

The authors have streamlined and clarified the manuscript. It is now suitable for publication in EMBO J.

MAX-PLANCK-INSTITUT für Multidisziplinäre Naturwissenschaften
KARL-FRIEDRICH-BONHOEFFER-INSTITUT

MPI für für Multidisziplinäre Naturwissenschaften, 37077 Göttingen

William Teale
Editor, EMBO J

Prof. Dr. Reinhard Lührmann
Zelluläre Biochemie

Am Faßberg 11 37077 Göttingen
Telefon: (0551) 201-1407
Telefax: (0551) 201-1197
E-Mail: reinhard.luehrmann@mpinat.mpg

January 25, 2024

Dear Dr. Teale,

In the newly revised version of our manuscript entitled “**New insights into the functions of B complex proteins revealed by cryo-EM of dimerized spliceosomes**” we have made the formatting changes that you requested in your letter. We also made changes/improvements as requested by Referee #1 as outlined below:

Referee #1:

The authors have responded to the referee comments and I think the manuscript is ready for publication. I just have a few minor things that the authors may want to consider at their own discretion.

1. I still find the possible connection to Ruth Sperling's work fascinating. The authors should note that given the symmetry of their structure it should easily be able to be expanded to a 4 spliceosome complex (but I think a 3 spliceosome complex would not be symmetry permitted) and that intramolecular formation of such a complex on the same (large) transcript should be favored over formation of it intermolecularly between different RNAs. In any case, I think it is worth a reference.

We now mention supraspliceosomes on p.7 and include a reference.

2. I don't like the idea of presenting data in favor of a conclusion that can't be shown or evaluated by reviewers. I would urge the authors to remove the sentence referring to the unpublished work on the MINX exon RNA substrate that lacked the 5' exon. I think the data is sufficient for showing the correct assignment.

We have deleted this sentence.

3. I still think it is misleading to propose that a spliceosome dimer may be phase separated in the absence of any supporting data. I suggest the following edit: "It is thus tempting to speculate that the diffuse globules that we observe in our hB dimers IN VITRO may CONTRIBUTE to bimolecular condensates containing exon RNA and, among others, SR proteins OBSERVED IN VIVO."

We changed this sentence according to the referee's suggestion.

4. I found Table S5 to be quite useful. It is still unclear to me why this information can't be represented in a (potentially quite useful) figure or video. Can the authors define what docking means for non structural biologists? In addition, a few things were "modeled for the first time...". Does this mean that they were previously docked and not modeled or were they "docked and modeled for the first time"?

We state more clearly what “docked” and “modeled for the first time” means in the legend to this table.

5. The cartoon figures in Appendix Figure S1 are so useful that they should be included in the main text figures. This would greatly facilitate the readability of the manuscript as well as increase the impact of the manuscript.

The cartoon figures shown in Appendix Figure S1 serve mainly as background information and for the most part do not fit well in the main figures. Thus we have chosen not to move them to the main figures.

We hope our manuscript is now suitable for publication in EMBO J.

Sincerely,

Reinhard Lührmann
Holger Stark

Dear Reinhard,

I am pleased to inform you that your manuscript has been accepted for publication in the EMBO Journal.

Congratulations on an important and elegant piece of work.

Best wishes,

William

William Teale, PhD
Editor
The EMBO Journal
w.teale@embojournal.org
